# Unraveling distinct effects between CuO$_x$ and PtCu alloy sites in Pt−Cu bimetallic catalysts for CO oxidation at different temperatures

Yunan Li[1,2,3,6], Lingling Guo[2,6], Meng Du[1,2], Chen Tian[1,3], Gui Zhao [4], Zhengwu Liu[1,3], Zhenye Liang[1,3], Kunming Hou[2], Junxiang Chen[5], Xi Liu [4], Luozhen Jiang[1,2] ✉, Bing Nan[1,2] ✉ & Lina Li[1,2] ✉

In situ exploration of the dynamic structure evolution of catalysts plays a key role in revealing reaction mechanisms and designing efficient catalysts. In this work, PtCu/MgO catalysts, synthesized via the co-impregnation method, out-performs monometallic Pt/MgO and Cu/MgO. Utilizing quasi/in-situ char-acterization techniques, it is discovered that there is an obvious structural evolution over PtCu/MgO from Pt$_x$Cu$_y$O$_z$ oxide cluster to PtCu alloy with sur-face CuO$_x$ species under different redox and CO oxidation reaction conditions. The synergistic effect between PtCu alloy and CuO$_x$ species enables good CO oxidation activity through the regulation of CO adsorption and O$_2$ dissocia-tion. At low temperatures, CO oxidation is predominantly catalyzed by surface CuO$_x$ species via the Mars-van Krevelen mechanism, in which CuO$_x$ can pro-vide abundant active oxygen species. As the reaction temperature increases, both surface CuO$_x$ species and PtCu alloy collaborate to activate gaseous oxygen, facilitating CO oxidation mainly through the Langmuir-Hinshelwood mechanism.

CO oxidation is a significant reaction in heterogeneous catalysis due to its practical applications in the purification of exhaust gases from motor vehicles, such as CO, NO, and hydrocarbons and as a typical model reaction for the fundamental catalysis study[1–3]. Supported platinum-based catalysts have attracted extensive attention recently due to their outstanding CO oxidation activity and stability at a relatively high temperature (>150 °C)[4,5]. However, the strong and oversaturated adsorption of CO on metallic Pt sites frequently inhibits CO oxidation at low temperatures because of CO poisoning and its poor ability to activate O$_2$[6]. With the increasingly stringent requirements of emission regulations, especially now the new standard for cold start emissions of vehicles, which requires cata-lysts to achieve CO oxidation conversion at low tempera-tures (<150 °C).

In the CO oxidation reaction, there are a lot of methods to improve the catalytic performance of Pt-based catalysts, such as con-structing alloy components, building metal-support interaction[7], metal-oxide[8], and metal-metal hydroxide[9] active sites. Among these methods, alloying Pt with another metal to form bimetallic alloy nanoparticles has been proven to be a promising pathway to improve the catalytic properties, such as Pt-Pd[10], Pt-Rh[11], Pt-Au[12], Pt-Ag[13], Pt-Fe[14], Pt-Co[15], Pt-Ni[16], and Pt-Cu[17–20] alloys. It is well known that elements with similar electronegativity are more likely to form alloys because of their easy share of electrons. Copper (Cu), as a representative transition metal, has received wide research due to its various valence states and rich surface oxygen species[21,22]. From the perspective of the alloy, Cu is a suitable element with a relatively narrow electronegativity gap and atomic radius difference with Pt atom, contributing to the substitution

[1]Shanghai Institute of Applied Physics, Chinese Academy of Sciences, Shanghai, China. [2]Shanghai Synchrotron Radiation Facility, Zhangjiang Laboratory, Shanghai Advanced Research Institute, Chinese Academy of Sciences, Shanghai, China. [3]University of Chinese Academy of Sciences, Beijing, China. [4]School of Chemistry and Chemical, In-situ Centre for Physical Sciences, Shanghai Jiao Tong University, Shanghai, China. [5]Division of China, TILON Group Technology Limited, Shanghai, China. [6]These authors contributed equally: Yunan Li, Lingling Guo. ✉e-mail: jianglz@sari.ac.cn; nanb@sari.ac.cn; lilina@sinap.ac.cn

in the lattice and avoiding significant lattice distortion[23]. Furthermore, PtCu alloys are also revealed as promising catalysts in CO oxidation due to low cost, special geometric, electronic, and multifunctional effects. Komatsu et al.[24] found that PtCu/SiO$_2$ alloy catalysts exhibited better CO oxidation activity because of electron transferring from Pt to Cu to promote the activation of O$_2$. Furthermore, Liu et al.[25] exploited that metallic Pt and Cu$^+$ in PtCu nanocage alloy could provide dual active sites for the adsorption of CO and O$_2$. In addition, PtCu alloy samples were employed as high-performance catalysts in other oxidation reactions, such as methanol oxidation[26], selective catalytic oxidation of ammonia[27], and polyhydric alcohol oxidation[28]. Although some detailed and profound researches on PtCu alloy have been investigated, there are still some debatable and unsolved problems for PtCu alloy in CO oxidation. Firstly, can the PtCu alloy be maintained during the whole CO oxidation? Secondly, what is the precise dynamic structural evolution of the PtCu alloy in the oxidation atmosphere, such as the oxidation level and the distinct role of Cu and Pt species in consideration of the instability and oxygen sensitivity of metallic Cu species? Thirdly, the lack of advanced in-situ characterization methods to explore what is the dominant mechanism (MvK or LH) in CO oxidation reaction for PtCu alloy catalysts with various active sites (alloy and oxide species)? The above existed problems matter the exploration and determination of active site and the "structure-activity" relationship in CO oxidation.

For supported catalysts, the choice of support frequently makes an influence on the catalytic performance through the formation of interaction between metal and support to improve the dispersion or regulate the electronic structure of the active site. MgO nanosheet offers a combination of high surface area, thermal stability, and cost-effectiveness, making it a favorable catalyst carrier for CO oxidation reaction[29,30]. Thus, in this work, MgO nanosheet was chosen as a good carrier to prepare platinum copper bimetallic catalysts by co-impregnation method, which displayed good performance compared to monometallic Pt/MgO or Cu/MgO catalysts. The in-situ XAFS results unravel the dynamic structural evolution of active site from platinum-copper oxide cluster to PtCu alloy-CuO$_x$ interface undergoing reductive and oxidized conditions. In situ DRIFTS/CO-TPR and isotope labeling experiments indicated that CO oxidation can be motivated at -50 °C on surface CuO$_x$ species through M-vK mechanism, in which CuO$_x$ can provide abundant active oxygen species. As the reaction temperature increases, a moderate CO adsorption on PtCu alloy guarantees enough sites for the activation of gas oxygen into active oxygen species to promote CO oxidation by L-H mechanism.

## Results
### Structural characterization of PtCu/MgO catalysts
MgO-supported platinum-copper catalysts were prepared by a co-incipient wetness impregnation method, denoted as PtCu/MgO, in which Pt/MgO and Cu/MgO catalysts were synthesized by the same method as reference. According to previous reports, the ratio of metal in bimetallic catalysts plays a key role in the catalytic performance in various reactions[31]. Therefore, the ratio of Pt and Cu is optimized by setting the content of platinum (0.5 wt.%) and regulating the amount of copper to acquire the best CO oxidation activity (Supplementary Fig. S1). It was found that when the designed content for copper was 6 wt.%, the best catalytic activity can be achieved. The ICP-AES results indicated that the actual content of platinum and copper (Pt: 0.48 and 0.54 wt.% for Pt/MgO and PtCu/MgO, respectively; Cu: 6.0, and 5.3 wt.% for Cu/MgO and PtCu/MgO, respectively) was well consistent with the designed value (Supplementary Table S1). In order to detect the basic structure information, the XRD experiment (Fig. 1a) was carried out. A typical MgO crystal (JCPDS 75-1525) can be observed for all samples, in which the diffraction peaks at 36.9°, 42.9°, 62.3°, 74.7°, and 78.6° were attributed to the (111), (200), (220), (311), and (222) planes of MgO, respectively. Moreover, no obvious diffraction peaks of Pt/

PtO/PtO$_2$ and Cu/Cu$_2$O/CuO were detected, presumably due to the highly dispersed platinum and copper species[32]. The morphology of the MgO support is nanosheets (Supplementary Fig. S2). In order to provide more microscopic data and clearly observe the precise distribution of Pt and Cu elements in PtCu/MgO, HAADF-STEM images and the related mapping were supplied in Fig. 1b–d. It can be seen that Pt and Cu elements were distributed uniformly at the cluster level within the same areas on the surface of MgO. For further confirmation about the elemental composition of clusters in PtCu/MgO, the corresponding line-profile analysis (Fig. 1d) manifest that Pt and Cu are homogeneously dispersed throughout the Pt$_x$Cu$_y$O$_z$ binary oxide clusters. In addition, because of the high loading of Cu species about 6 wt.%, a part of Cu species existed in isolation without forming interaction with Pt species according to the reduction peak at 209 °C (CuO$_x$ cluster) in H$_2$-TPR profile of PtCu/MgO (Fig. 1e). Therefore, CuO$_x$ and Pt$_x$Cu$_y$O$_z$ clusters coexisted in PtCu/MgO. Because of the high dispersion and low contrast between Cu ($Z$ = 29) and Mg ($Z$ = 12), it is difficult to visually observe CuO$_x$ clusters over Cu/MgO in STEM image (Supplementary Fig. S3). The observed cluster in PtCu/MgO (Fig. 1b) can be attributed to Pt$_x$Cu$_y$O$_z$ binary oxide clusters with an average particle size of 1.4 ± 0.3 nm. More importantly, the H$_2$-TPR results (Fig. 1e) also uncovered the strong interaction between platinum and copper in PtCu/MgO. A strong reduction peak at 187 °C assigned to Pt-[O]$_x$-Cu structure was detected in PtCu/MgO[33], which was different from the reduction peaks of PtO$_x$ (213 °C) and CuO$_x$ (209 °C) species. It indicated that the interaction between platinum and copper could significantly enhance the reducibility of catalysts. The XAFS spectra further provide more precise electronic and coordination information about copper and platinum. The Pt L$_3$-edge XANES profiles in Fig. 1f showed that the white line intensity of Pt/MgO was higher than that of PtCu/MgO, in which the average oxidation state of Pt for Pt/MgO and PtCu/MgO was 4 and 3.8, respectively (Supplementary Table S2). According to the fitting results of EXAFS of PtCu/MgO in Fig. 1g and Supplementary Table S2, the strong Pt-O ($R$ ≈ 1.98 Å and CN ≈ 5.5), Pt-O-Cu ($R$ ≈ 3.10 Å and CN ≈ 2.8), and Pt-O-O shells were acquired, further evidencing the formation of Pt$_x$Cu$_y$O$_z$ component. The absence of Pt-O-Pt shells suggests that there is no isolated PtO$_x$ cluster in PtCu/MgO. In addition, the coordination structure of Cu species was also investigated. In Fig. 1h, the Cu K-edge XANES spectra of Cu/MgO and PtCu/MgO are very similar in both peak position and line shape. The average oxidation state of Cu/MgO and PtCu/MgO were both +2. For PtCu/MgO, it is a challenge to fit the Cu-O-Pt coordination shell at Cu K edge, because XAFS is a bulk technique sensitive to all of the forms of an element in a sample. In practical, XAFS experiment, millions of x-rays are absorbed by millions of atoms, and the average information of one element was collected together, such as Cu-O-Cu (CuO$_x$) and Cu-O-Pt (Pt$_x$Cu$_y$O$_z$) in this system. In this work, the isolated CuO$_x$ species is much more abundant than Pt$_x$Cu$_y$O$_z$ clusters, which results in the dominated CuO coordination structure (Fig. 1i). A main Cu-O shell ($R$ ≈ 1.97 Å, CN ≈ 2.0 and $R$ ≈ 1.96 Å, CN ≈ 2.0) plus Cu-O-Mg shell ($R$ ≈ 2.95 Å, CN ≈ 4.7 and $R$ ≈ 2.93 Å, CN ≈ 5.4) can be fitted for both Cu/MgO and PtCu/MgO samples (Supplementary Table S3). A minor Cu-O-Cu shell ($R$ ≈ 2.84 Å, CN ≈ 0.5 and $R$ ≈ 2.81 Å, CN ≈ 0.6) can be fitted for Cu/MgO and PtCu/MgO, which indicates that there are very small clusters of copper oxide on Cu/MgO and PtCu/MgO. Therefore, a combination of TEM, H$_2$-TPR, and XAFS results, it evidenced the formation of Pt$_x$Cu$_y$O$_z$ binary oxide cluster (Fig. 1j) in PtCu/MgO possessing different reducibility and structure from PtO$_x$ and CuO$_x$ clusters.

### Catalytic performance of PtCu/MgO catalysts in CO oxidation
CO oxidation has been widely used as a model reaction for the fundamental study of reaction mechanisms and the surface properties of catalysts. According to previous reports, the pretreatment conditions have a profound effect on the catalytic performance via regulating the oxidation state or the local coordination structure of active site in

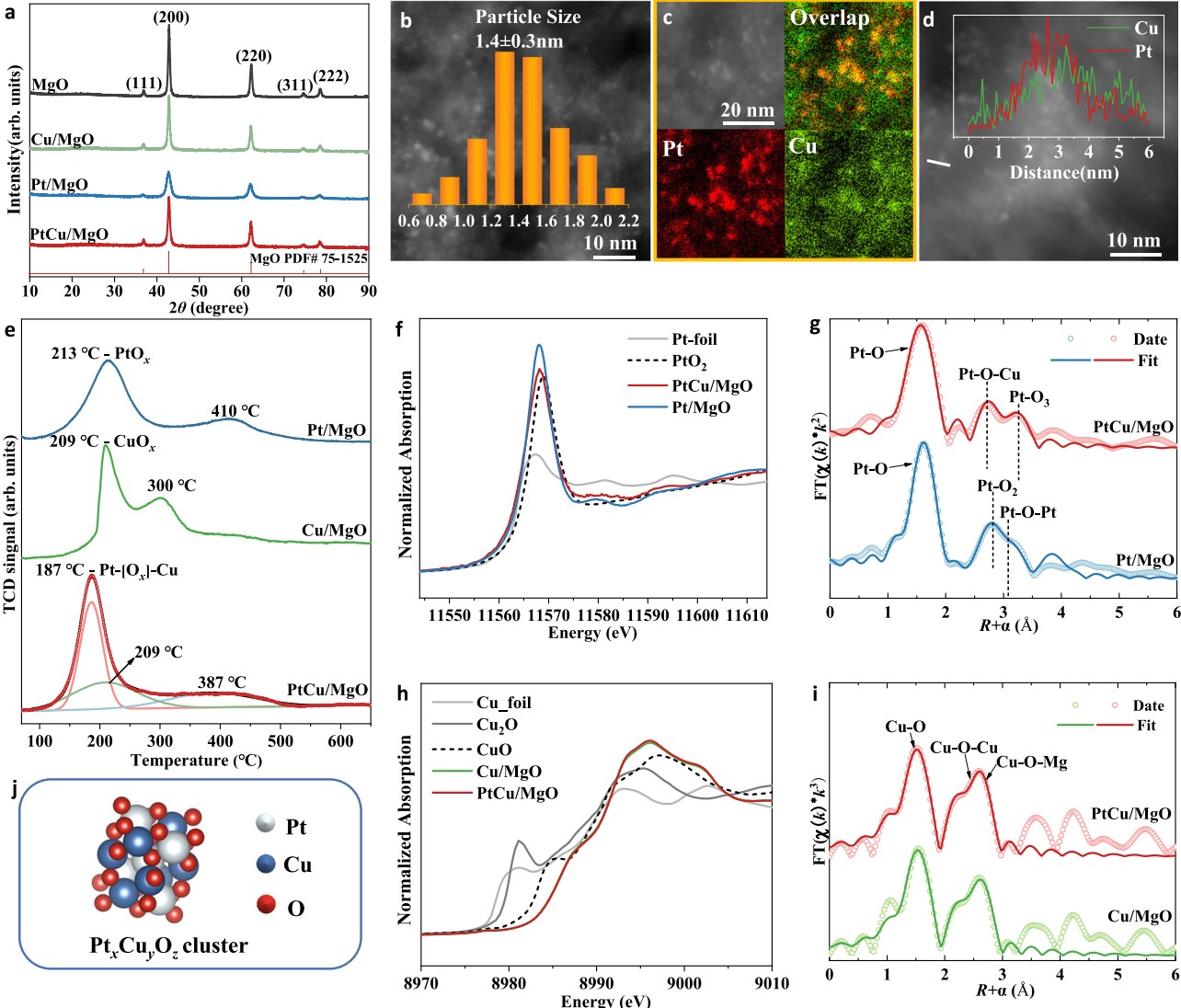

**Fig. 1 | Structural characterization of PtCu/MgO catalysts. a** XRD pattern, (**b**) HAADF-STEM images of PtCu/MgO, (c) EDS mapping results of PtCu/MgO, (d) line-scanning results of PtCu/MgO, (**e**) H₂-TPR profiles, (**f**) Pt L₃-edge XANES profiles, (**g**) Pt L₃-edge EXAFS fitting results (the data are $k^2$-weighted and not phase-corrected), (**h**) Cu K-edge XANES profiles, (**i**) Cu K-edge EXAFS fitting results in R space (the data are $k^3$-weighted and not phase-corrected), (**j**) Schematic demonstration of platinum-copper oxide cluster.

various bimetallic catalysis systems[31]. In this work, when the catalysts were pretreated under $O_2$ atmosphere, all samples exhibited bad CO oxidation activity with a complete conversion temperature of CO oxidation >200 °C (Fig. 2a), presumably due to the poor adsorption of CO molecules on active site, according to the only existence of gas CO bands in in-situ DRIFTS results (Supplementary Fig. S4). However, there is a dramatic improvement in activity for all samples with hydrogen reduction at 500 °C. PtCu/MgO exhibited good CO oxidation activity with 100% CO conversion at 130 °C, even can motivate this reaction at −50 °C (Fig. 2b). For monometallic Pt and Cu samples, the CO complete conversion can only be achieved at 154 °C and 248 °C with few activity below 100 °C. It demonstrated that the hydrogen reduction may not only reduce the oxidation state of platinum and copper species, but also regulate the coordination structure of active site. In order to verify our assumption, the related kinetic data was acquired (Fig. 2c). The apparent activation energy ($E_a$) of the PtCu/MgO catalyst is around 42 kJ·mol⁻¹, which is much lower than that of Pt/MgO (82 kJ·mol⁻¹), Cu/MgO (90 kJ·mol⁻¹) and other reported $E_a$ values in Supplementary Table S4 (from 45 to 98 kJ·mol⁻¹) of Pt-based[34–36] or Cu-based catalysts[22,37], indicating the difference in active structure or

reaction pathway. Besides, a long-term catalytic evaluation at 150 °C for PtCu/MgO under a simulative vehicle exhaust condition (GHSV:190,000 ml h⁻¹ $g_{cat}$⁻¹) showed 94% conversion without any deactivation within 20 h (Fig. 2d). Meanwhile, pure Pt catalyst also showed good stability at 150 °C with a poor conversion efficiency of about 20% under the same test conditions. In addition, PtCu/MgO also exhibited good catalytic stability under water vapor condition (Supplementary Fig. S5). Moreover, PtCu/MgO showed better catalytic activity compared to other platinum-based catalyst in Supplementary Table S5.

**Structural evolution of PtCu/MgO after hydrogen reduction**
According to our previous research, the enhancement of hydrogen reduction on catalytic activity is frequently accompanied by the evolution of active structure[31]. Therefore, in order to acquire more precise structural information under reduction conditions, the catalysts were deeply characterized. After hydrogen reduction, the size of the active site in PtCu/MgO was maintained at ~1.8 nm (Fig. 3a), in which the particle of PtCu/MgO in Fig. 3b can be assigned to PtCu alloy (CuPt JCPDS 48-1549) with a clear lattice spacing of 0.22 nm for the (111)

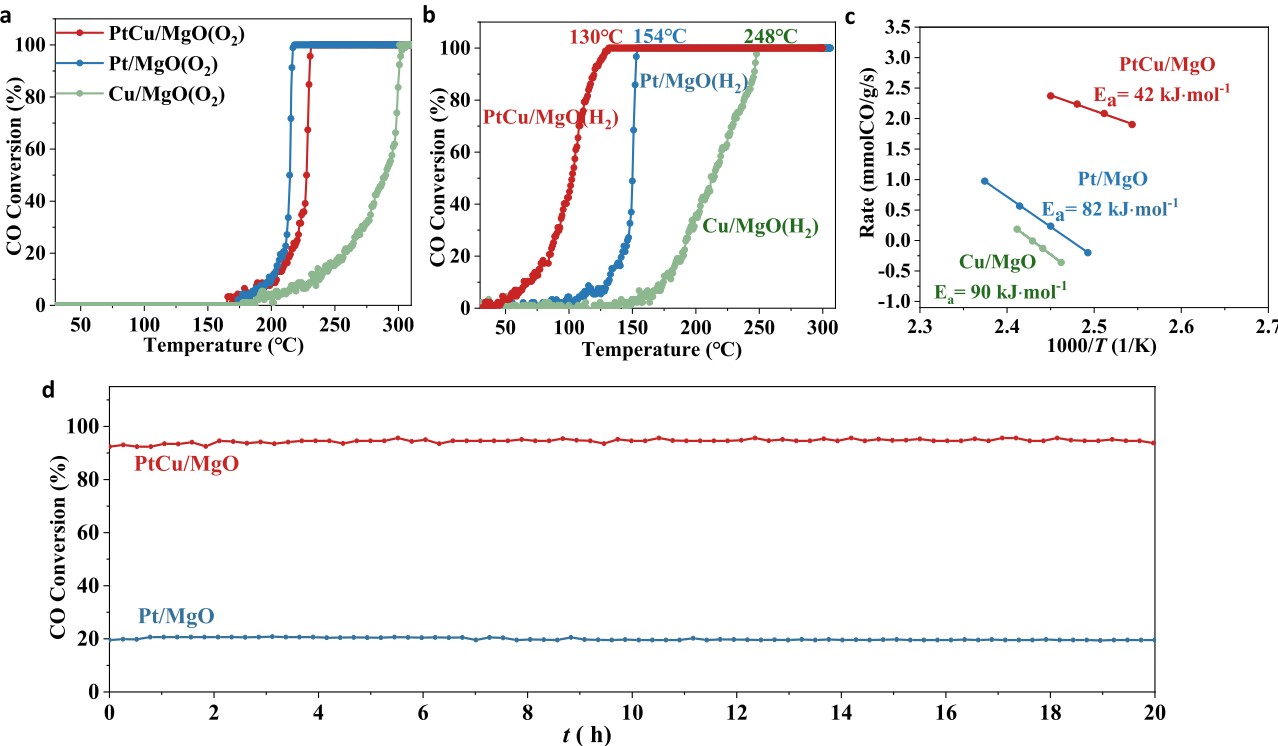

**Fig. 2 | Catalytic performance of PtCu/MgO catalysts in CO oxidation. a, b** Catalytic performance for all catalysts (1vol.%CO/20%O$_2$/79%He, 120,000 ml h$^{-1}$ g$_{cat}$$^{-1}$), (**c**) Arrhenius plots, (**d**) Stability test of PtCu/MgO and Pt/MgO at 150 °C for CO oxidation reaction (GHSV: 190,000 ml h$^{-1}$ g$_{cat}$$^{-1}$).

plane. Meanwhile, the related EDS mapping results also evidenced the space consistency of Pt and Cu element distribution (Fig. 3c). Zhang et al. also prepared ultrasmall Pt-Cu alloy clusters on the surface of TiO$_2$ NBs after hydrogen reduction at 400 °C for 2 h[38]. In order to further confirm the formation of PtCu alloy, quasi-situ synchrotron XRD and in-situ XAFS were conducted. The XRD pattern for PtCu/MgO after hydrogen reduction exhibited two additional peaks positioned at 40.1°, 50.4° (Fig. 3e), which match with the (111) planes of PtCu alloy and the (200) planes of metallic Cu, respectively. The in-situ XANES in Supplementary Fig. S6, and Fig. S7 show that the average oxidation states of Cu and Pt were both 0 valence in PtCu/MgO after hydrogen reduction. The in-situ EXAFS of Pt L$_3$-edge for PtCu/MgO(H$_2$) exhibits one prominent peak at -2.60 Å (Fig. 3f) with a coordination number of ~11.8 (Supplementary Table S6), which is significantly different from the Pt foil, and is in accordance with the standard path of Pt-Cu from CuPt (mp-644311 in The Materials Project)[39]. The in-situ EXAFS of Cu K-edge for PtCu/MgO(H$_2$) in Fig. 3g exhibits one prominent peak of Cu-Cu shell at -2.59 Å (CN ≈ 6.3) (Supplementary Table S7 and Supplementary Fig. S15), which indicates that copper oxide clusters were reduced into metallic Cu particles after hydrogen pretreatment. To further strengthen the formation of PtCu alloy, wavelet transforms (WT) analysis of Pt EXAFS oscillations was conducted. The WT contour plots of Pt foil (Fig. 3h) and PtO$_2$ (Supplementary Fig. S8) demonstrate that the intensity maxima at -10 Å$^{-1}$ and -5 Å$^{-1}$ are attributed to the Pt-Pt and Pt-O contributions, respectively. However, for the WT contour plot of PtCu/MgO (H$_2$) (Fig. 3i), just one intensity maximum at about 7 Å$^{-1}$ is shown, which is attributed to the Pt-Cu contribution[40]. Therefore, the results of HAADF-STEM, quasi in-situ synchrotron XRD and in situ XAFS characterizations verify the formation of PtCu alloy in PtCu/MgO after H$_2$ reduction (Fig. 3d).

### Structural evolution during CO oxidation

Due to the oxidation atmosphere during CO oxidation reaction, the coordination structure of the active site, especially the oxidation state, may undergo further evolution. After CO oxidation, the aberration-

corrected HAADF-STEM images showed that the average particle sizes of the active site for PtCu/MgO (used) (Fig. 4a), Pt/MgO (used) (Supplementary Fig. S9) were 1.9 ± 0.6 nm, 1.7 ± 0.4 nm, respectively. The high-resolution TEM (HRTEM) images of PtCu/MgO used (Supplementary Fig. S9) display a typical lattice fringe of Pt-Cu alloy with d-spacing of 0.22 nm from (111) plane. The corresponding inverse Fourier transfer (IFFT) pattern and the line intensity profile also show an interplanar space of 0.22 nm. Meanwhile, the line scan results of PtCu alloy particles in Fig. 4b obviously evidenced the element distribution of Pt and Cu in PtCu alloy. It demonstrated that PtCu alloy structure is stable during the whole CO oxidation reaction. The quasi in-situ synchrotron XRD pattern of PtCu/MgO(used) in Fig. 4c also shows the diffraction signal at 40.1° from (111) crystal plane of PtCu alloy, confirming the existence of PtCu alloy in PtCu/MgO after CO oxidation reaction. For MgO-supported Pt sample (Supplementary Fig. S9), the main active site is metallic platinum with 0.23 nm for Pt (111) plane. Furthermore, the structure-sensitive in-situ XAFS technique was employed to determine the precise structural evolution of active site for PtCu/MgO in Fig. 4d–i. During the CO oxidation reaction at 90 °C, the Pt-Cu shell remained stable and without obvious the formation of Pt-O shell (Fig. 4e and Supplementary Table S6). Figure 4d and Supplementary Fig. S7a show that the white line peak gradually increased with the increase in reaction temperature, indicating the slight oxidation of platinum species during CO oxidation reaction with +0.8 and +1.8 state for Pt species at 150 °C and 270 °C respectively (Fig. 4f and Supplementary Fig. S7b). Meanwhile, the XPS results in Supplementary Fig. S10 also evidenced that the platinum species underwent oxidation after CO oxidation. However, the main PtCu alloy was still stabilized during CO oxidation reaction at <150 °C (Fig. 4e and Supplementary Table S6). For Pt/MgO catalysts after CO oxidation reaction, a major metallic Pt-Pt shell (R ≈ 2.77 Å, CN ≈ 6.0) plus Pt-O shell (R ≈ 2.02 Å, CN ≈ 1.7) can be fitted (Supplementary Fig S11 and Table S2), indicating the existence of PtO$_x$ species was not the dominant factor in promoted CO oxidation activity. For Cu species in PtCu/MgO, it can be reduced to metallic Cu species with zero

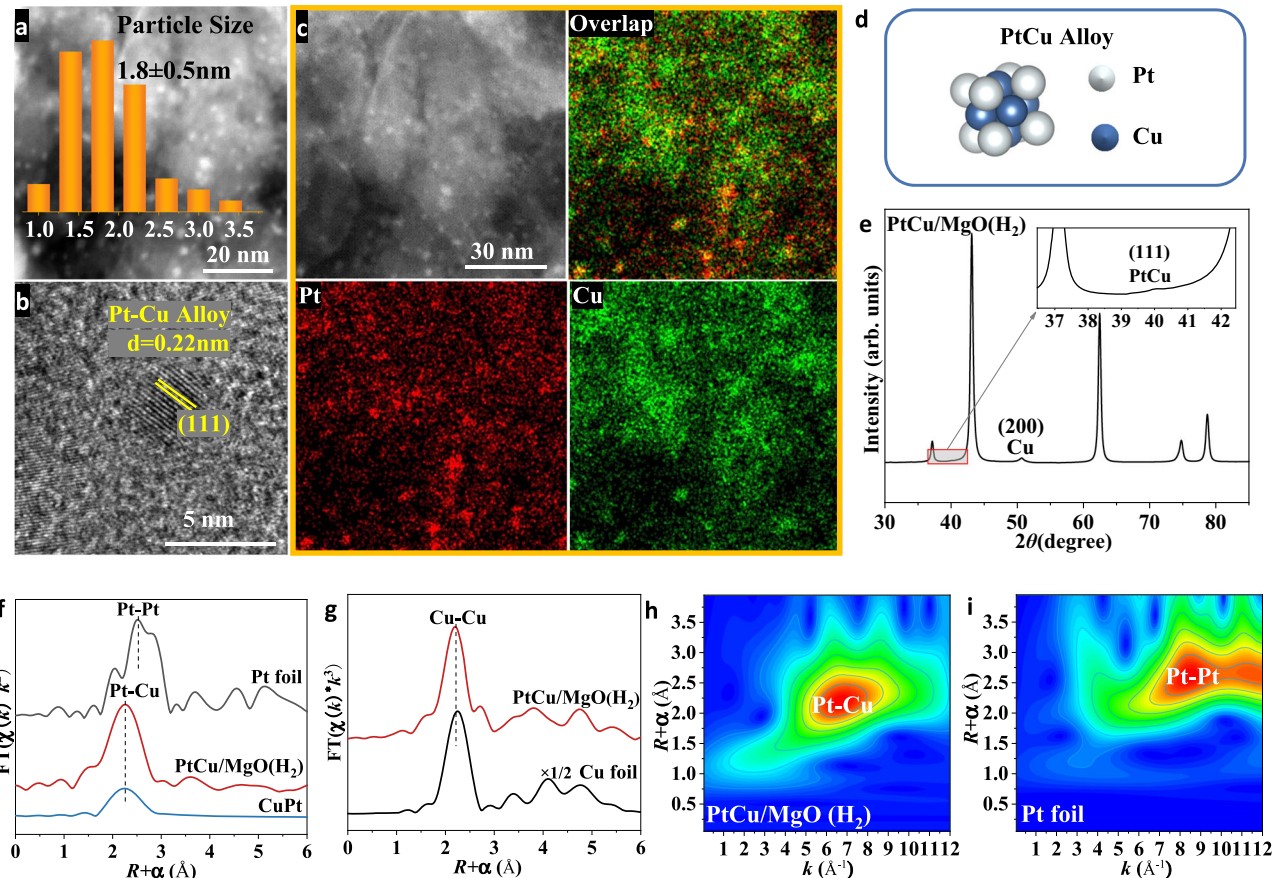

**Fig. 3 | Structural characterization of PtCu/MgO after hydrogen reduction.** **a** HAADF-STEM images, (**b**) HRTEM, (**c**) EDS mapping results of PtCu/MgO after hydrogen pretreatment, (**d**) Schematic illustration of PtCu Alloy, (**e**) synchrotron XRD graph, (**f**) in-situ Pt L$_3$-edge EXAFS profiles (the data are $k^2$-weighted and not phase-corrected), (**g**) in-situ Cu K-edge EXAFS profiles (the data are $k^3$-weighted and not phase-corrected), and (**h**, **i**) WT-EXAFS contour plot of Pt L$_3$-edge signals for Pt foil and the PtCu/MgO after hydrogen reduction.

valence after hydrogen reduction at 500 °C. When it switched to CO oxidation condition at 90 °C, Cu-O and Cu-O-Cu shells began to appear, indicating the formation of CuO$_x$ species (Fig. 4h). During the CO oxidation reaction at 150 °C, the metallic Cu-Cu shell completely disappeared with an oxidation state of +0.4, indicating that the reoxidation and redispersion of isolated Cu species and Cu species on the surface of PtCu alloy. For comparison, the XAFS data for Cu/MgO were also acquired in Supplementary Fig. S12. The isolated CuO$_x$ species can be reduced to metallic Cu component after hydrogen reduction and remained at metallic state (Cu$^0$) during the CO oxidation at 90 °C. It indicated that in PtCu/MgO sample after hydrogen activation, when the reaction gas was switched into the reactor, the Cu atoms in PtCu alloy could rapidly disassociate O$_2$ gas into oxygen species, together with the formation of CuO$_x$ species on the surface of PtCu alloy. Furthermore, the metallic Cu particles can be seen in Cu/MgO (used) sample by the aberration-corrected HAADF-STEM (Supplementary Fig. S9), and there is still small Cu-Cu shell (CN ≈ 0.7) can be fitted in Cu/MgO (used) (Supplementary Fig. S13 and Table S3), indicating that the complete dispersion of the isolated metallic Cu particles into small CuO$_x$ clusters requires a higher temperature in Cu/MgO. Tomita et al. also reported that the formation of the interface between metallic Pt nanoparticles and FeO$_x$ promoted the oxidation of CO at low temperatures[14]. Furthermore, the aberration-corrected HAADF-STEM images of PtCu/MgO could also display some clusters at subnanometer on the surface of PtCu alloy, which may be assigned to CuO$_x$ cluster on the basis of element contrast (Supplementary Fig. S9c). Therefore, it can be concluded that the main active site for PtCu/MgO during CO oxidation is PtCu alloy with surface CuO$_x$ species (Fig. 4j). In addition,

we found that there is a slight deactivation after multiple cycles of CO oxidation (Supplementary Fig. S14), which may be due to the degradation of PtCu alloy under long time operation in oxidation atmosphere. However, the activity can be recovered to initial state by H$_2$ reduction.

## The reducibility and active oxygen for PtCu/MgO catalysts
We also carried out the in-situ H$_2$-TPR experiments (Fig. 5a) for the samples after CO oxidation without air exposure to detect the reducibility of catalysts. No visual peak can be found for used Pt/MgO due to the vast majority of platinum species is reduced to Pt$^0$ during H$_2$ pretreatment and the oxidation state of platinum is insensitive to oxidative atmosphere. However, for Cu/MgO and PtCu/MgO, there is an obvious reduction peak at 170 or 155 °C, which was attributed to the reduction of well dispersed copper oxide species. It indicated that there were abundant CuO$_x$ species on PtCu alloy, which is more reducible than the conventional CuO$_x$ species in Cu/MgO. Meanwhile, the XANES at Cu K edge (Supplementary Fig. S6) and XPS results (Fig. 5b) uncovered that the average oxidation state of Cu on PtCu/MgO catalyst surface was +2 after CO oxidation. Moreover, the in-situ CO-TPR results in Fig. 5c also uncovered the difference in surface active oxygen species between PtCu/MgO and Cu/MgO. The surface oxygen in PtCu/MgO can react with CO molecules at -50 °C, well consistent with the CO oxidation "light off" temperature (Fig. 2b). The CuO$_x$ species on the surface of PtCu alloy may motivate initial CO oxidation (-50 °C) through Mars-van Krevelen (M-vK) mechanism[31], promoting the CO oxidation activity. Nan et al. also found super active oxygen species in Pt-based catalysts to exhibit excellent CO oxidation

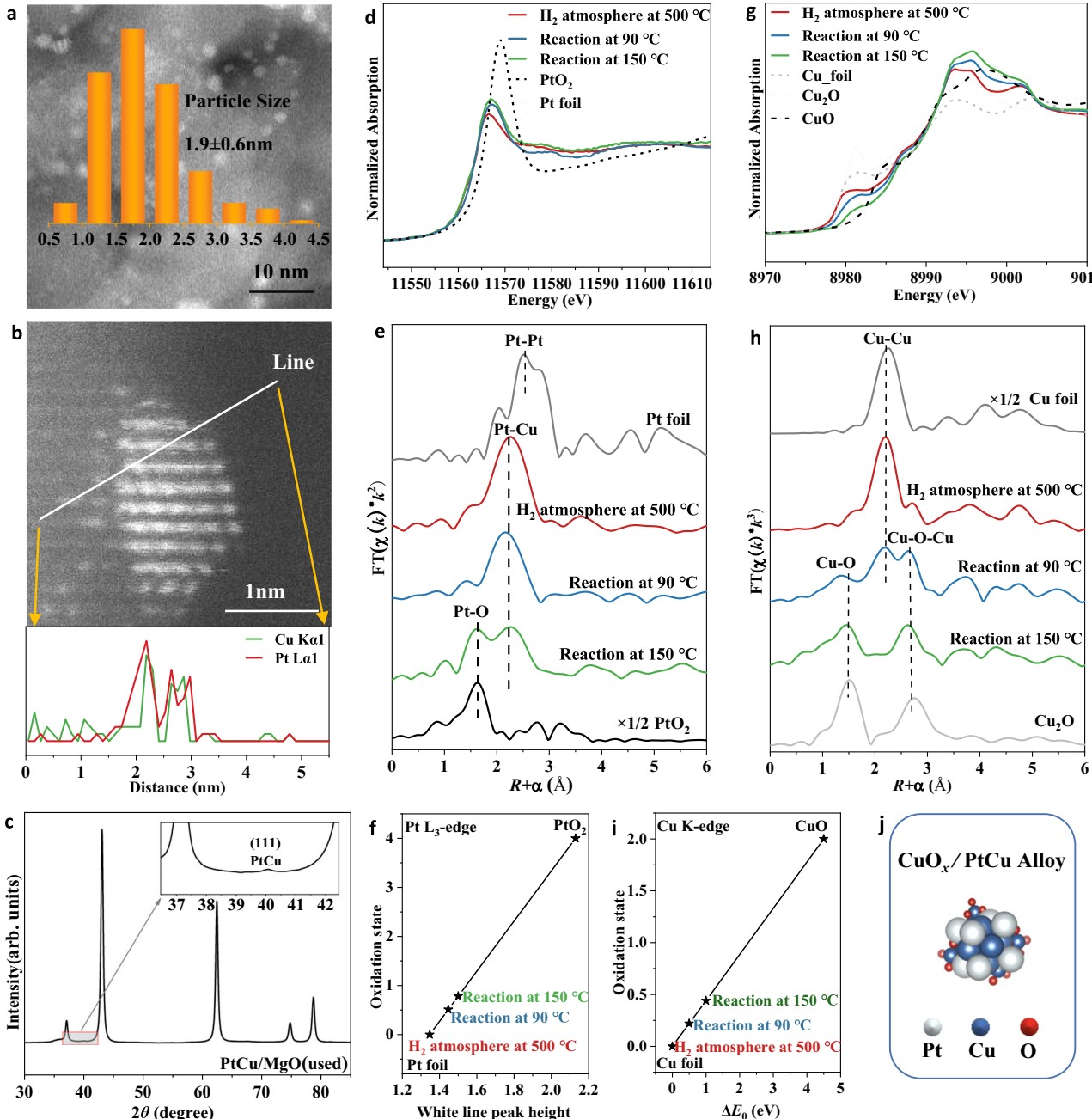

**Fig. 4 | Structural characterization of PtCu/MgO catalysts during CO oxidation.** **a** aberration-corrected HAADF-STEM images of PtCu/MgO after catalytic CO oxidation, (**b**) the line scan results of PtCu/MgO after catalytic CO oxidation, (**c**) synchrotron XRD pattern, (**d**) in-situ Pt $L_3$-edge XANES profiles, (**e**) in-situ Pt $L_3$-edge EXAFS profiles (the data are $k^2$-weighted and not phase-corrected), (**f**) the average oxidation state of Pt in PtCu/MgO catalysts from XANES spectra, (**g**) in-situ Cu K-edge XANES profiles, (**h**) in-situ Cu K-edge EXAFS profiles (the data are $k^3$-weighted and not phase-corrected), (**i**) the average oxidation state of Cu in PtCu/MgO catalysts from XANES spectra, and (**j**) Schematic illustration of $CuO_x$/PtCu Alloy.

activity[31]. However, the formation of $CO_2$ was only detected over Pt/MgO and Cu/MgO at much higher temperatures (>150 °C). It indicated that the isolated $CuO_x$ species make little contribution to CO oxidation activity at low temperatures. Moreover, the latter peak in the range 230–350 °C is attributed to the water-gas shift reaction of surface hydroxyl groups bounded to MgO. In addition, the EPR results in Fig. 5d exhibited symmetric signals at g of about 2.003, attributed to electrons trapped on the $O_v$[41]. It was found that PtCu/MgO possessed the most abundant oxygen defects possibly stemming from the $CuO_x$ species on the surface of PtCu alloy.

## The ability of oxygen activation

$O_2$-TPD experiment was performed to characterize the active oxygen species on the catalyst surface. According to $O_2$-TPD results (Supplementary Fig. S16), a distinct desorption peak at the range of 100–200 °C can be ascribed to the surface-adsorbed oxygen species. Surface-adsorbed oxygen species desorbed at low temperatures are known as the active species for catalytic oxidation. The desorption temperature for PtCu/MgO is lower than that of Pt/MgO and Cu/MgO, indicating that the adsorbed oxygen is easier to be activated in PtCu/MgO than in Pt/MgO and Cu/MgO. In addition, we designed an $O_2$

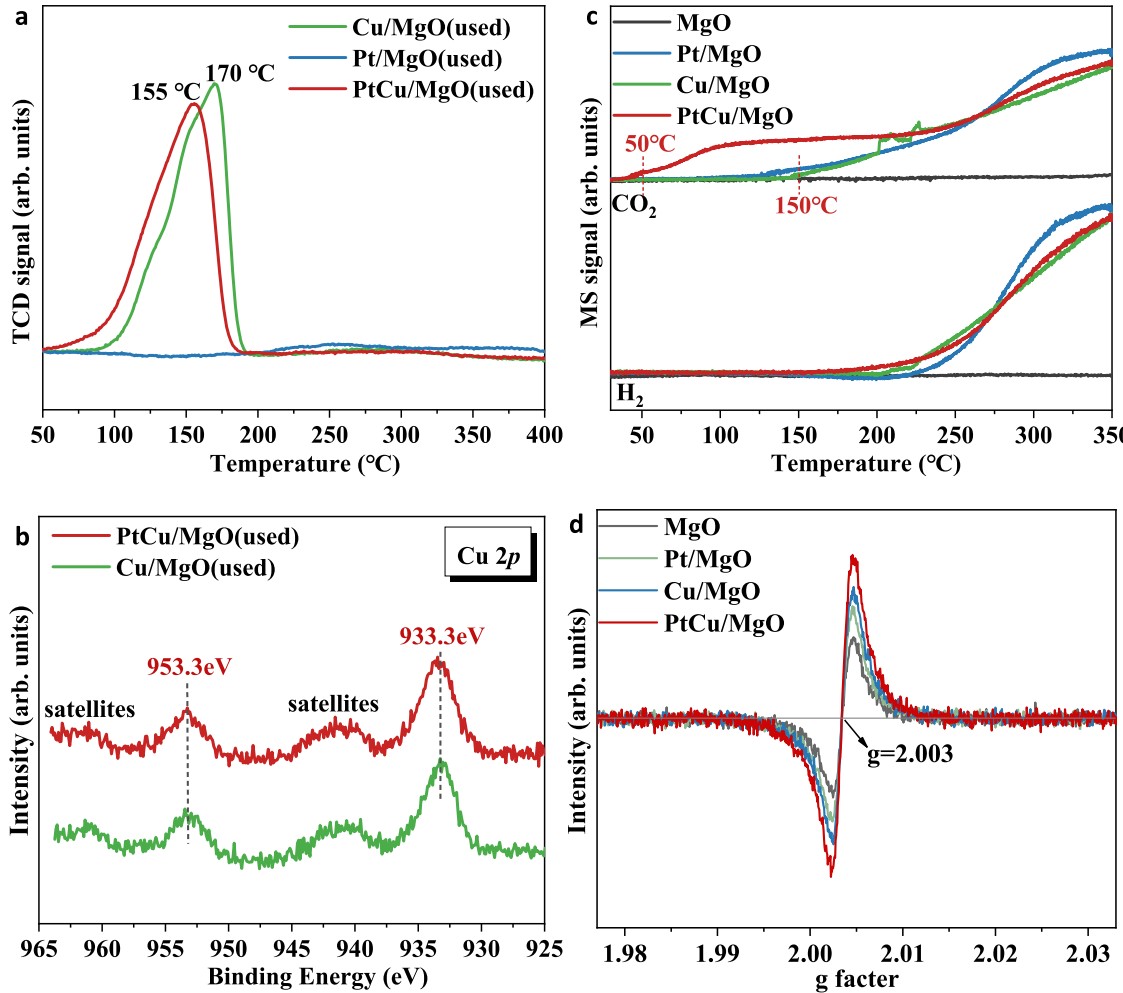

**Fig. 5 | The reducibility and active oxygen for PtCu/MgO catalysts. a** In-situ $H_2$-TPR profiles for used PtCu/MgO catalysts after CO oxidation without contact to air, (**b**) XPS spectra of Cu $2p$ for used PtCu/MgO catalysts, (**c**) CO-TPR patterns of PtCu/MgO catalysts, (**d**) EPR spectra of used PtCu/MgO catalysts.

pulse experiment to test the ability of the oxygen decomposition of each catalyst in Fig. 6. At 70 °C, PtCu/MgO can activate $O_2$ to generate $CO_2$, while no obvious $CO_2$ signal can be seen for Pt/MgO and Cu/MgO due to their poor ability to dissociate oxygen gas. As the experiment temperature increased to 130 °C, the integral area of $CO_2$ signal for PtCu/MgO catalyst was larger than that for Pt/MgO and Cu/MgO, indicating a better decomposition capacity of $O_2$ for PtCu/MgO. In addition, it was found that there was a $CO_2$ shoulder peak in PtCu/MgO, which was not consistent with the variation tendency of oxygen gas. It can be attributed to the consumption of oxygen species in $CuO_x$ species with good recyclability on the surface of PtCu alloy. For Pt/MgO and Cu/MgO, the generated $CO_2$ signal of Pt/MgO was much better than that of Cu/MgO, well consistent with the trend of apparent activation energy. When the $O_2$ pulse experiment is conducted at 170 °C, there is also additional $CO_2$ generation for Cu/MgO. However, the peak area of Cu/MgO was smaller than that of PtCu/MgO, indicating that $CuO_x$ on the surface of PtCu alloy has better oxygen activation ability than isolated Cu species.

## CO adsorption on PtCu/MgO catalysts
The characterization results from synchrotron XRD, STEM, in situ-XAFS, and TPR revealed the exact structure of active sites for Pt/MgO (metallic Pt), Cu/MgO ($CuO_x$), and PtCu/MgO (PtCu alloy with surface $CuO_x$). Furthermore, the adsorption behaviors of reaction molecule also play a key role in catalytic performance. Thus, an in-situ DRIFTS

experiment was carried out to investigate the specific adsorption and desorption patterns. As shown in Fig. 7a, the CO adsorption bands at 2174 and 2115 $cm^{-1}$ were attributed to gaseous CO[31,42]. In addition, the peaks in Pt/MgO (Supplementary Fig. S17) at 2064 and 2082 $cm^{-1}$ were attributed to linear CO adsorbed on $Pt^0$, $Pt^{\delta+}$ sites, respectively[43,44]. The CO adsorption peaks in Cu/MgO (Supplementary Fig. S17) at 2086 $cm^{-1}$, 2103 $cm^{-1}$, and 2134 $cm^{-1}$ were attributed to linear CO adsorbed on $Cu^0$, $Cu^+$, and $Cu^{2+}$ sites, respectively[2,45]. According to the in-situ EXAFS fitting results, PtCu alloy with surface $CuO_x$ species is the main active site for PtCu/MgO. Meanwhile, the CO adsorption bands at 2086, 2106, 2134 $cm^{-1}$ (CO-$Cu^0$, CO-$Cu^{+1}$ and CO-$Cu^{2+}$) and 2064 $cm^{-1}$ (CO-$Pt^0$) were clearly observed in the spectra of PtCu/MgO, confirming the reliability of structural characterization. During CO adsorption process for PtCu/MgO, CO molecules adsorbed at $Pt^0$ site (2064 $cm^{-1}$) quickly reached saturation at 120 s, the CO adsorption on the surface $CuO_x$ cluster (2106 $cm^{-1}$) also quickly reached saturation at 120 s and the peak intensity is the highest, indicating the rich $CuO_x$ cluster on the surface of PtCu alloy. In addition, there was a tiny band at 2013 $cm^{-1}$, which was attributed to the CO adsorption on PtCu alloy[15,46]. Later, when 2 vol.% $O_2/N_2$ was introduced into the cell, it can be seen that the bands at 2106 and 2134 $cm^{-1}$ (CO-$Cu^{1+}$ and CO-$Cu^{2+}$) rapidly decreased and the peaks at 2064 $cm^{-1}$ remained at the beginning stage with gradual diminution after 200 s. It indicated that the $CuO_x$ species on the surface of PtCu alloy was good at the dissociation of $O_2$ into reactive O radicals to convert CO to $CO_2$. However, the strong

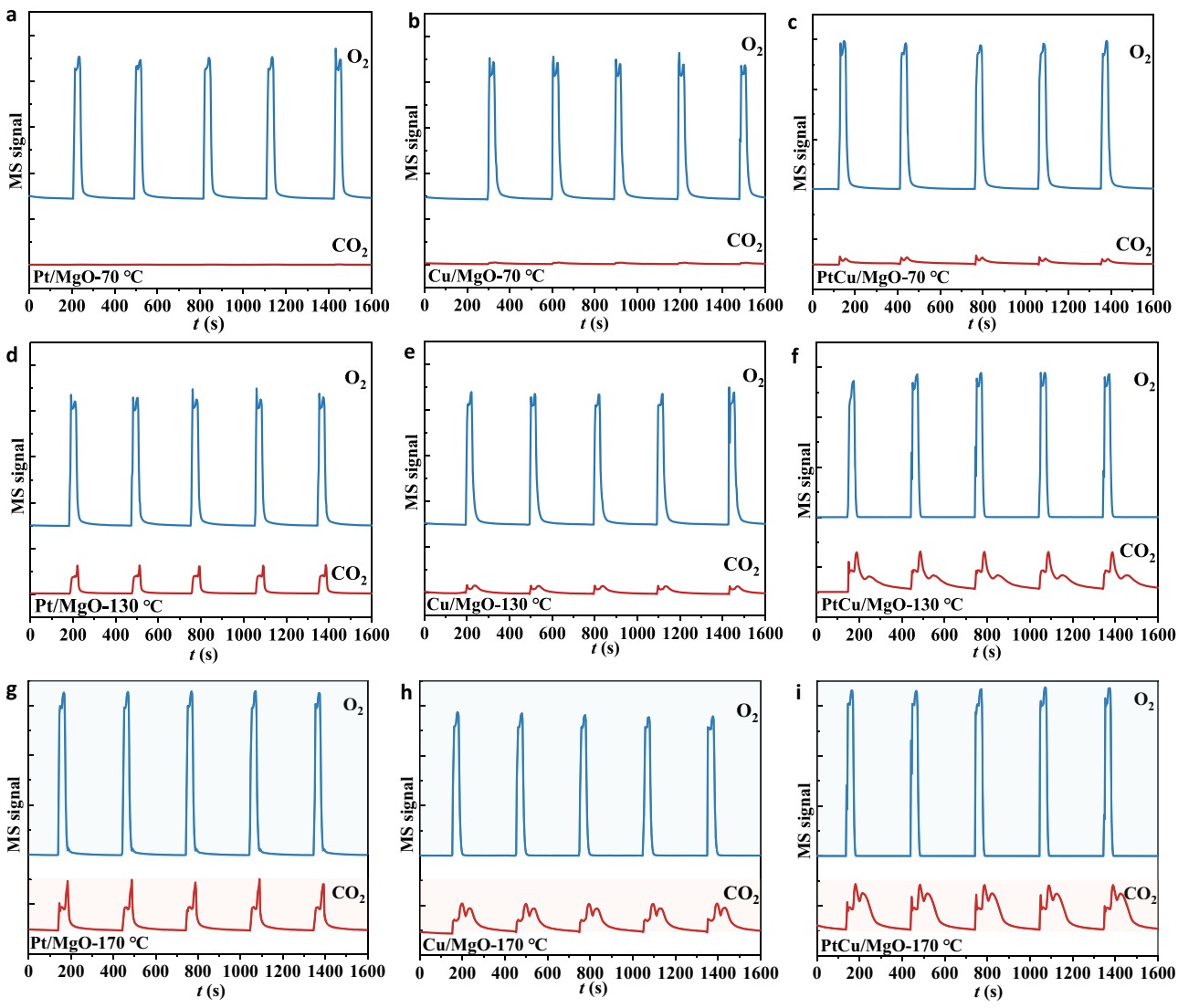

**Fig. 6 | The ability of oxygen activation.** $O_2$ pulse experiment for Pt/MgO (**a**, **d**, **g**), Cu/MgO (**b**, **e**, **h**) and PtCu/MgO (**c**, **f**, **i**) catalysts at different temperature.

adsorption on $Pt^0$ species resulted in not enough sites to efficiently activate $O_2$ gas and motivate further CO oxidation. In Fig. 2b, the light off profile of PtCu/MgO also exhibited the good CO oxidation activity at low temperatures (<100 °C) for two reasons: the abundant active oxygen species in $CuO_x$ and the better ability to dissociate $O_2$ molecules. Therefore, in order to detect the role of PtCu alloy in CO oxidation activity, we conducted further DRIFTS experiment at relatively high temperatures (200 °C) in Fig. 7c. It can be seen that there was a visible variation for the CO adsorption behavior of PtCu/MgO. The adsorption peak strength of CO on $Cu^+$ (2016 $cm^{-1}$) and $Pt^0$ (2064 $cm^{-1}$) decreased, while the adsorption peak strength of CO on PtCu alloy (2013 $cm^{-1}$) significantly increased, indicating that CO molecules prefer to adsorb on PtCu alloy rather than $CuO_x$ species and $Pt^0$ sites at relatively high temperatures. More importantly, the CO adsorbed at the Pt sites saturated quickly (40 s), while the CO adsorbed slowly on the PtCu alloy with unsaturation until 1600 s. Thus, the slow adsorption of CO at the PtCu alloy site provides enough sites for the activation of gas oxygen into active oxygen species. When the gas $O_2$ was introduced into PtCu/MgO system (Fig. 7d), the peaks at 2106 and 2013 $cm^{-1}$ simultaneously and rapidly vanished at 160 s, which indicates that the high temperature can provide sufficient energy for both $CuO_x$ and PtCu alloy to activate the gas oxygen to participate in CO oxidation.

## Reaction mechanism of PtCu/MgO catalysts

According to in-situ DRIFTS results, $CuO_x$ species and PtCu alloy play a different role in CO oxidation activity at different temperature ranges. The CO oxidation can be motivated by $CuO_x$ species because of their abundant active oxygen species at low temperatures (<100 °C). As the reaction temperature increased, PtCu alloy efficiently adsorbed CO molecules and dissociated gas $O_2$ to promote the conversion of CO. For the CO oxidation reaction, two possible mechanisms, e.g., the LH and M-vK mechanisms, have been proposed. According to the $^{18}O_2$ isotope-labeling results over PtCu/MgO (Fig. 7e, f), the M-vK and LH reaction mechanisms played a predominant role in different temperature ranges, respectively. It can be seen that when the reaction temperature is 70 °C, surface $CuO_x$ species could provide active oxygen species to react with CO molecules and generate $C^{16}O_2$. Meanwhile, the $O_v$ in $CuO_x$ could dissociate gaseous $^{18}O_2$ to produce active $O^{18}$ radicals and promote the further generation of $C^{16}O^{18}O$. On the basis of MS signal of $CO_2$, the M-vK mechanism occupied a dominate position (~63%) in CO oxidation at 70 °C. When the reaction temperature is 150 °C, with the increment of CO adsorption at the PtCu alloy site and the easier dissociation of gaseous $O_2$ at this site, the catalyst can quickly convert CO molecules and $^{18}O_2$ into $C^{16}O^{18}O$ through the LH mechanism. At this temperature, the CO oxidation was mainly promoted by LH mechanism (~60%) in Fig. 7f. In addition, the

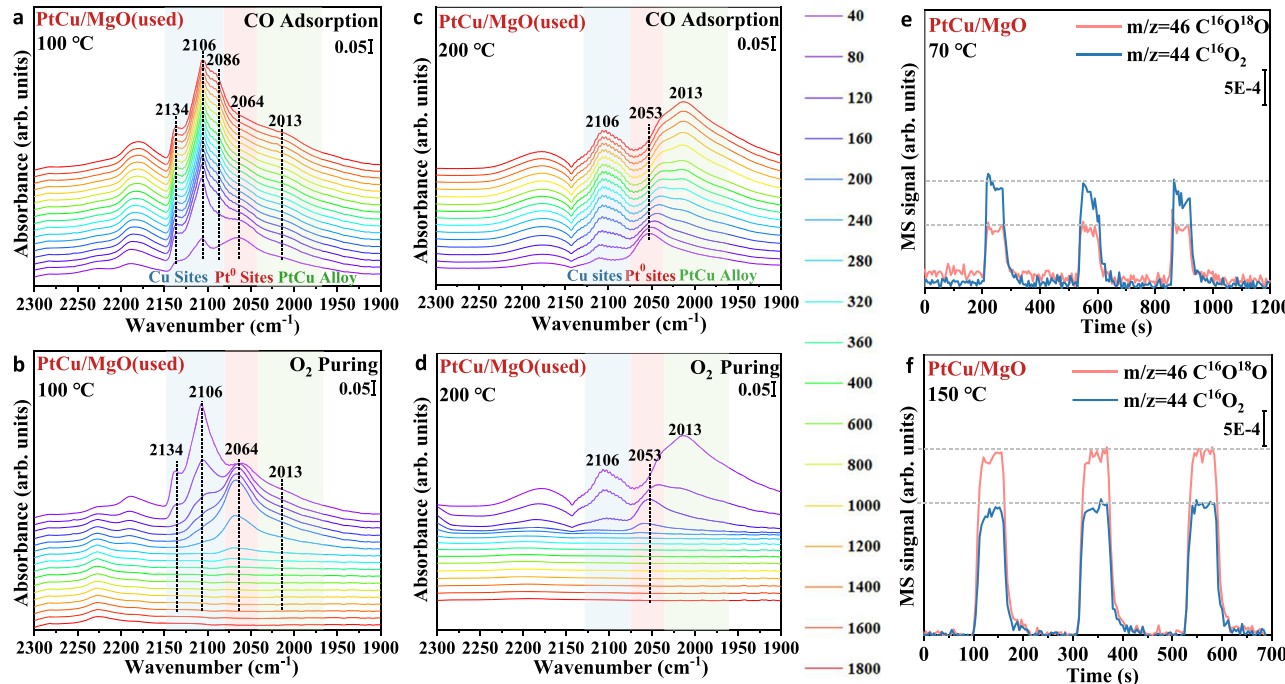

**Fig. 7 | CO adsorption and $^{18}O_2$ isotope-labeling experiments for PtCu/MgO catalysts at different temperature. a–d** In-situ DRIFTS spectra for used PtCu/MgO catalysts tested at 100 °C and 200 °C (The catalysts were pretreated at 500 °C for 30 min under $H_2$ flow and underwent CO oxidation reaction from 30 to 300 °C in the DRIFTS reaction cell before data collection) and (**e, f**) $^{18}O_2$ isotope-labeling experiments.

related $E_a$ of PtCu/MgO at low temperatures (38 kJ·mol$^{-1}$) is also slightly lower than that (42 kJ·mol$^{-1}$) at high temperatures (Supplementary Fig. S18) because of the different roles of CuO$_x$ species and PtCu alloy in CO oxidation. Thus, it can be seen that the CO molecules prefer to adsorb on surface CuO$_x$ species to drive the occurrence of CO oxidation with the affluent and active oxygen species provided by CuO$_x$ component at low temperatures. As the reaction temperature increases, it can supply enough energy to meet the energy requirement of PtCu alloy to participate in CO oxidation, plus the strong and slow CO adsorption and the efficient activation of $O_2$ over PtCu alloy (Fig. 8b).

## Discussion

To sum up, MgO-supported Pt-Cu bimetallic catalysts were prepared by the co-impregnation method, and displayed good activity compared to the related monometallic Pt/MgO or Cu/MgO catalysts. The apparent activation energy ($E_a$) of the PtCu/MgO catalyst is around 42 kJ·mol$^{-1}$, which is lower than that of Pt/MgO (82 kJ·mol$^{-1}$) and Cu/MgO (90 kJ·mol$^{-1}$). Combined with various characterization methods including in-situ XAFS, quasi in-situ synchrotron XRD, and STEM, we reveal an obvious structural evolution over PtCu/MgO catalysts (Fig. 8a), in which from Pt$_x$Cu$_y$O$_z$ binary oxide cluster (air calcination) to PtCu alloy ($H_2$ activation) and PtCu alloy with surface CuO$_x$ species (main active site in CO oxidation). The simultaneous existence of the PtCu alloy and CuO$_x$ species enables a synergy for catalyzing CO oxidation. It is discovered that PtCu alloy can activate $O_2$ gas to form surface CuO$_x$ species and provide oxygen species to motivate CO oxidation by M-vK mechanism at low temperatures. At high temperatures, both CuO$_x$ and PtCu alloy work together to activate gas oxygen to participate in CO oxidation. Around 60% of the acquired $CO_2$ is produced by O atoms from the introduced $O_2$ gas adsorbed on PtCu alloy and CuO$_x$ (L-H mechanism).

## Methods
### Catalyst preparation

All of the chemicals were obtained from Sinopharm Chemical Reagent Co. Ltd. and used without additional purification. MgO is synthesized by thermal decomposition. Firstly, 0.04 mol Mg(CH$_3$COO)$_2$·4H$_2$O and 0.06 mol H$_2$C$_2$O$_4$·2H$_2$O were dissolved in 50 ml and 200 ml Millipore (>18 MΩ) water, respectively. Under constant agitation, the dissolved magnesium acetate solution was dropped into oxalic acid solution. At this time, precipitation was not immediately generated. After 12 h aging at room temperature, the final pH value of the solution was 2. Then, the precipitates were washed by Millipore (>18 MΩ) water for three times. The MgO was obtained by drying the as-washed product under a vacuum at 80 °C for 12 h and further calcined in air at 600 °C for 10 h (heating rate: 2 °C min$^{-1}$). Deposition of platinum, and copper onto the MgO support was carried out by a co-incipient wetness impregnation. Firstly, a solution of Cu (NO$_3$)$_2$·3H$_2$O (0.24 g) and Pt (NH$_3$)$_4$(NO$_3$)$_2$ (0.01 g) in Millipore (>18 MΩ) water (5 ml) was added dropwise onto MgO power (1 g) under manually stirring. Subsequently, the sample was dried at 80 °C for 12 h, and calcined at 400 °C for 4 h (ramping rate: 2 °C min$^{-1}$).

### Catalytic activity tests

The CO oxidation activities of MgO-supported platinum-copper catalysts were conducted in a fixed-bed flow reactor. 30 mg of catalysts were first pretreated under 5 vol.% H$_2$/He (30 mL·min$^{-1}$) at 500 °C for 30 min. After cooling down to room temperature under H$_2$ atmosphere, the atmosphere was switched to 1 vol.% CO/20 vol.% O$_2$/He (60 mL·min$^{-1}$). Afterward, the temperature was ramped to 300 °C at 5 °C min$^{-1}$. Nondispersive IR spectroscopy (Gasboard-3000, Hubei Ruiyi Company, China) was used to continuously monitor the compositions of CO and CO$_2$ in the output gases. The conversion of CO was evaluated as follows: CO$_{conv.}$ (%) = (CO$_{in}$−CO$_{out}$)/CO$_{in}$ × 100%.

### Materials characterization

Inductively coupled plasma optical emission spectroscopy (ICP-OES, Optima 8000) was used to determine the metal content for all sample. The powder X-ray diffraction (XRD) patterns were collected on a Bruker D8 Advance diffractometer (40 kV and 40 mA) using the Cu $K_a$ radiation ($k = 1.5418$ Å) from 10 to 90°. N$_2$ adsorption/desorption isotherms were measured on an ASAP2020-HD88 analyzer (Micromeritics

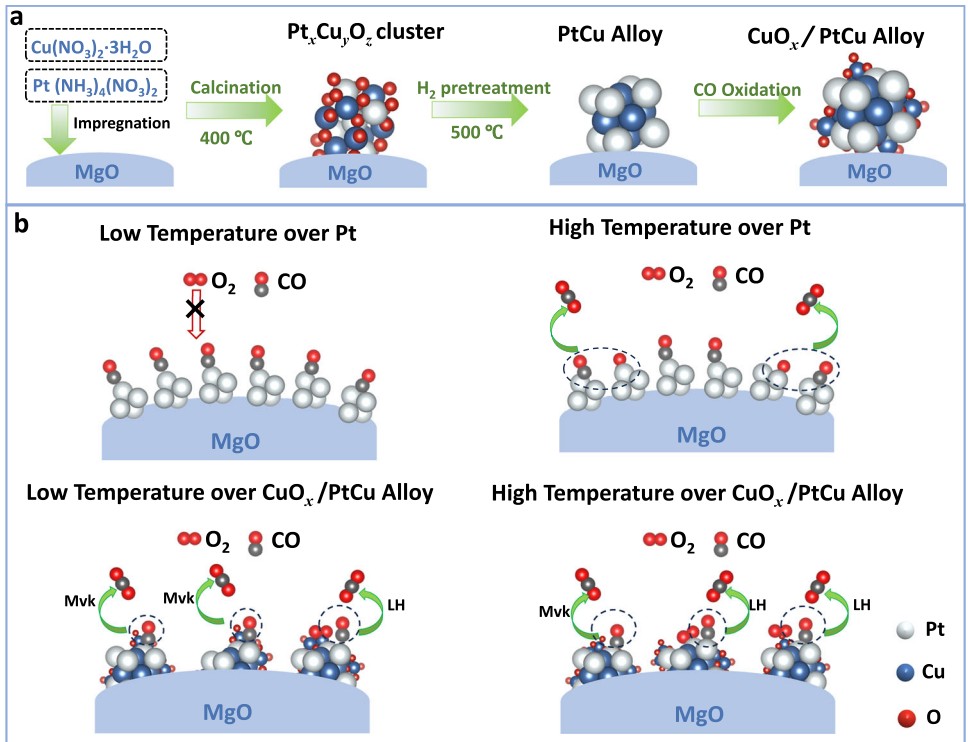

**Fig. 8 | Schematic illustration of structural evolution and reaction mechanisms. a** Schematic illustration of structural evolution for the PtCu/MgO catalyst during $H_2$ pretreatment and CO oxidation reaction, (**b**) Graphical representation of proposed reaction mechanisms for CO oxidation over Pt at low temperatures, Pt at higher temperature, $CuO_x$/PtCu alloy at low temperatures, and $CuO_x$/PtCu alloy at higher temperature.

Co., Ltd.) at 77 K. The samples were degassed at 250 °C for 4 h. The BET specific surface areas ($S_{BET}$) were calculated using data between 0.05 and 0.20 relative pressure. The transmission electron microscopy (TEM) and scanning transmission electron microscopy-energy dispersive spectrometer (STEM-EDS) elemental mapping results were obtained from a FEI Tecnai G2 F20 S-TWIN microscope and transmission electron microscopy of cold field emission (JEOL JEM-F200) operating at 200 kV. The scanning electron microscope (SEM) images were taken on a field emission scanning electron microscope (Zeiss, G300). The aberration-corrected HAADF-STEM images and EDS mappings were obtained using a probe aberration-corrected Hitachi HF5000 equipped with a secondary detector, operated at 200 kV. X-ray photoelectron spectroscopy (XPS) was performed by PHI 5000 VersaProbe III with a monochromatic Al Kα X-ray source. Rectify the binding energy for all the spectra using the C 1 s peak at 284.8 eV as an internal standard. The quasi in-situ synchrotron XRD pattern were recorded at BL17B of the Shanghai Synchrotron Radiation Facility (SSRF) at a wavelength of 0.68883 Å. The catalysts were pretreated in a stainless reactor with two air valves on each side. After the sample is cooled to room temperature, the stainless reaction tube with catalyst is transferred to a glove box under $N_2$ atmosphere to prepare sample for the test. Pilatus3 S-2M detector was used to record the diffraction Pattern. $LaB_6$ standard was used for wavelength calibration. The 2θ Angle was converted to the corresponding value of the Cu $K_a$ radiation ($k = 1.5418$ Å) by Scherrer formula.

## In-situ/ex-situ XAFS experiments
The X-ray absorption fine structure (XAFS) spectroscopy at the Pt $L_3$ ($E_0 = 11564$ eV) edge and Cu K ($E_0 = 8979$ eV) edge were operated at the BL14W and 16U1 beamline of Shanghai Synchrotron Radiation Facility (SSRF) which operates at 3.5 GeV with a current of 200 mA. To investigate chemical coordination environment of Pt species and Cu species for PtCu/MgO catalysts after $H_2$ activation, during, and after the CO oxidation reaction, the in-situ XAFS experiment were carried out. The

50 mg of catalyst was placed on a sample holder in the center of a high-temperature high-pressure gas-solid catalytic in-situ device. Firstly, the sample was pretreated with $H_2$ at 500 °C for 30 min, followed by cooling to 30 °C under $H_2$. Then, the XAFS spectra of Pt $L_3$ edge and Cu K edge were collected under $H_2$ atmosphere. Subsequently, 1vol.%CO/20%$O_2$/79%He was introduced into the in-situ device, when the reaction temperature reached 90 °C, 150 °C, 210 °C, and 270 °C, the XAFS spectra were collected. For the quasi in-situ XAFS experiment, firstly, the catalysts were pretreated in a stainless reactor with two air valves on each side under $H_2$ flows at 500 °C for 30 min. After the sample is cooled to room temperature, the stainless reaction tube with catalyst is transferred to a glove box under $N_2$ atmosphere to prepare sample for XAFS test. The XAFS spectra of Pt $L_3$ edge and Cu K edge were collected in fluorescence mode using an Ar-filled Lytle detector. The X-ray energy was calibrated with the absorption edge of Pt foil and Cu foil. The data were analyzed using the Demeter software package (including Athena and Artemis software)[47]. Athena was used for data normalization data preprocessing and Artemis was used for EXAFS fitting.

## In-situ DRIFTS
In situ diffuse reflectance infrared Fourier transform spectroscopy (in situ DRIFTS) were continuously collected using a Thermo Nicolet iS10 spectrometer with a mercury-cadmium-telluride (MCT) detector cooled with liquid nitrogen. The 6 mg of catalyst was placed on a sample holder in the center of diffuse reflectance cell (Harrick system). Prior to the testing, the sample was pretreated with $H_2$ at 500 °C for 30 min, followed by cooling to 30 °C under purging with He gas. Then, the sample is heated and kept at 100 °C. The background spectra were acquired under He purging. Subsequently, 5 vol.% CO/He (20 mL min⁻¹) was introduced into the in-situ cell for CO adsorption for 30 min. This was followed by He gas purging and CO re-adsorption for 30 min. Afterward, 5 vol.% $O_2$/He was introduced.

## Reducible property and surface oxygen

Hydrogen temperature program reduction ($H_2$-TPR) experiments were performed on Micromeritics Autochem II 2920 instrument equipped with a thermal conductivity detector TCD detector. 50 mg of the catalysts was placed in a U-type quartz tube. The catalysts were pretreated at 300 °C for 30 min under $O_2$ (5 vol.%$O_2$/He) flow before the analysis, and then cooled to room temperature. Subsequently, 5 vol.% $H_2$/Ar was introduced for analysis. After stabilizing the TCD signal in $H_2$ flow (50 mL min$^{-1}$) at room temperature, the reduction temperature was raised from room temperature to 700 °C at rate of 10 °C min$^{-1}$.

CO-temperature programmed reduction (CO-TPR) experiment was conducted in a fixed-bed flow reactor connected to online mass spectrometry (LC-D200M, TILON). 50 mg of catalysts were first pretreated under 5 vol.% $H_2$/He (30 mL min$^{-1}$) at 500 °C for 30 min. Afte the sample was purged by He and then cooled down to room temperature, the atmosphere was switched to 2.2 vol.% CO/He (60 mL min$^{-1}$). Afterward, the temperature was ramped to 400 °C at 10 °C min$^{-1}$. During the heating process, the signals of $H_2$ (m/z = 2), He (m/z = 4) and $CO_2$ (m/z = 44) were recorded.

Electron paramagnetic resonance (EPR) measurements were performed at the X-band using a Bruker Emxplus spectrometer. The center field was 3505.00 G, the microwave frequency was 9.843 GHz, the microwave power was 3.170 mW.

## $O_2$ pulse experiment

$O_2$ pulse experiments were performed in a fixed-bed flow reactor connected to online mass spectrometry (LC-D200M, TILON). The catalysts were firstly treated under 5%$H_2$/He atmosphere at 500 °C for 30 min. Afterward, the catalyst is then cooled to room temperature in $H_2$ atmosphere. Then the sample was heated from room temperature to 70 °C in He atmosphere, 2%CO/He (30 mL min$^{-1}$) was then introduced to purging until the baseline is stable. Subsequently, $O_2$ (20 vol.% $O_2$/He) pulses of 7.5 mL were injected. The signals of $CO_2$ (m/z = 44), $O_2$ (m/z = 32) were detected. Because of the total reduction of Pt and Cu after $H_2$ activation, the production of $CO_2$ is from the decomposition of oxygen gas.

## $^{18}O_2$ isotope-labeling experiments

$^{18}O_2$ isotope-labeling experiments were performed in a fixed-bed flow reactor connected to online mass spectrometry (LC-D200M, TILON). 30 mg of catalysts were first activated in 5 vol.% $H_2$/He (30 mL min$^{-1}$) at 500 °C for 30 min. Afterward, the catalyst is then cooled to room temperature in $N_2$ atmosphere. Then the sample was heated from room temperature to 70 °C, $^{18}O_2$ was then introduced to purging until the baseline is stable. Subsequently, CO (2.2 vol.% CO/He) pulses of 15 cm$^3$ were injected at 100 s intervals. The signals of $C^{16}O_2$ (m/z = 44), $C^{16}O^{18}O$ (m/z = 46) were detected.

## Data availability

The data that support the findings of this study are available within the paper and its Supplementary Information. The data generated in this study are provided in the Source Data file. Source data are provided with this paper.

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

## Acknowledgements

This work was supported by "Photon Science Research Center for Carbon Dioxide", Science and Technology Innovation Plan of Shanghai Science and Technology Innovation Commission (23YF1453700 received by B.N.), "Shanghai Science and Technology Innovation Action Plan" (22JC1403800 received by L.N.L), "2022 Self Deployed Instrument Design Project of Shanghai Advanced Research Institute" and "National Natural Science Foundation of China"(12105351 received by B.N.). We appreciate the assistance of TILON Group Technology Limited (Division of China) and In-situ Center for Physical Sciences, Shanghai Jiao Tong University in characterization of catalysts. Additionally, the User Experiment Assist System of SSRF, CAS-Shanghai Science Research Center, and 14 W, 13SSW, 06B, 17B and 16U1 beamline of SSRF provided support for data collecting for this work.

## Author contributions

L.N.L. supervised the work and procured funds. Y.N.L., L.L.G., L.Z.J., and B.N. designed the experiments and analyzed the results. M.D. assisted in-situ DRIFTS data analysis. C.T. and Z.Y.L. carried out XRD and XPS experiments. G.Z. and X.L. performed the aberration-corrected HAADF-STEM measurements. L.L.G. carried out the TEM/HRTEM measurements. Y.N.L. and L.Z.J. performed the $^{18}O_2$ isotope-labeling experiments. Y.N.L., Z.W.L., and K.M.H. carried out XAS measurements and analyzed the data. J.X.C. assisted with the TPR test. Y.N.L. and B.N. wrote the manuscript. All authors discussed the results and revised the manuscript.

## Competing interests

The authors declare no competing interests.
