## [Peer Review File · Nature Communications]

Unraveling the Distinct Effects between CuOx and PtCu alloy Sites in Pt–Cu Bimetallic Catalysts for CO Oxidation at Different TemperaturesREVIEWER COMMENTS

Reviewer #1 (Remarks to the Author):

The manuscript from Yunan Li et al. addresses the CO oxidation reaction mechanism on the Pt-Cu-MgO powder catalyst. By means of XAS, TEM and H₂-TPR, the PtCu catalyst was characterized. Further combined with reactivity data, MS and FT-IR, the CO oxidation reaction mechanism was resolved, which was dependent on the reaction temperature. The MvK reaction mechanism prefers low temperatures in contrast to the L-H mechanism occurring at higher temperatures. The new mechanistic insights into the small PtCu clusters towards CO oxidation reaction and also its predominant reactivity contribute to the research on the advanced catalyst. Although, the manuscript is well written, the results and discussions are not clearly presented. Therefore, I suggest that this manuscript should be suitable for publication in Nature Communications after major revision.

Reviewer #2 (Remarks to the Author):

The authors conducted an investigation into PtCu/MgO catalysts for CO oxidation, employing a series of quasi in-situ characterization experiments to elucidate the catalyst's structural changes under reductive and oxidized conditions. Additionally, they explored the reaction mechanism of PtCu-CuOx catalysts across varying reaction temperatures. While the reported data represents a comprehensive investigation, the reviewer has discerned that this work, rather than ushering in novel findings, makes an incremental contribution. Consequently, this manuscript lacks the level of novelty for publication in a prestigious journal like Nature Communications. The PtCu/CuOx catalytic system, the identification of the combined MvK and LH mechanism, and the utilization of approaches such as isotopic labeling are not original. I would suggest its submission to a journal in a more specified field.

In addition to the references cited within the manuscript, it is prudent to consider the following related studies:

"Catalytic activity of PtCu intermetallic compound for CO oxidation: A theoretical insight," published in Catalysis Today, 383, 2022, 339-344.

"Dual Active Sites of CO Adsorption and Activation over PtCu Alloy Nanocages for Preferential Oxidation of Carbon Monoxide," featured in ACS Applied Energy Materials, 2022, 5(1), 604-614.

"Ordered intermetallic Pt-Cu nanoparticles for the catalytic CO oxidation reaction," presented in RSC Advances, 2016, 6, 85634-85642.

"PtCu Intermetallic Compound Supported on Alumina Active for Preferential Oxidation of CO in Hydrogen," found in The Journal of Physical Chemistry C, 2013, 117(20), 10483-10491.

Reviewer #3 (Remarks to the Author):

This manuscript has been employed a simple co-impregnation method to prepare PtCu/MgO catalysts. By utilizing various characterization techniques, it was observed that under reduction and oxidation conditions, platinum-copper oxide cluster underwent significant restructuring into PtCu alloy-CuOx interfaces. The synergistic effect between PtCu alloy and CuOx clusters endowed the catalyst with excellent CO oxidation activity. In situ DRIFTS/CO-TPR and isotope labeling experiments indicated that surface CuOx species at low temperatures could stimulate CO oxidation via the M-vK mechanism, while rising to higher temperatures, CO oxidation followed the L-H mechanism. This work holds certain research significance, particularly regarding the structural evolution from platinum-copper oxide cluster to alloy and alloy with cluster, providing important insights into active sites. However, the manuscript also exhibits notable shortcomings. For example, the innovative contributions of the manuscript are not explicitly delineated, and there is a lack of more direct evidence for the characterization of active sites. Additionally, many analytical aspects within the manuscript suffer from internal contradictions. Consequently, as outlined below, several important aspects need addressing before recommending this article for publication in Nature Communications. Specific suggestions are as follows:

1. The Introduction lacked logical coherence. While the manuscript discussed Pt-Cu alloy and CuOx interfaces as active sites for CO oxidation, the authors did not sufficiently elaborate on the scientific basis for choosing Pt and Cu as the research subjects in the introduction, especially in explaining the scientific rationale behind the formation of Pt and Cu alloy. Additionally, the manuscript failed to summarize the research progress and existing issues related to PtCu alloy catalysts in CO oxidation. The current manuscript form did not adequately demonstrate the value of studying the PtCu alloy catalyst system. Furthermore, the rationale for selecting the carrier should be explained in the introduction. Lastly, the third paragraph of the mechanism explanation was overly lengthy and should be appropriately condensed.
2. In Figure 1e, the H₂ reduction peaks of PtCu/MgO catalyst overlapped with the reduction peaks of Cu/MgO and Pt/MgO catalysts. However, the authors attributed this solely to Pt-[Ox]-Ce species. How can the presence of CuOx or PtOx species be ruled out?
3. In Figure 2d, the authors compared the PtCu/MgO catalyst with Cu/CoFe₂O₄, Pt₃Ti/SiO₂, and Pt/CuCrOx catalysts, which are evidently different from the PtCu/MgO catalyst discussed in the manuscript. It is suggested that the authors choose more representative catalysts for the comparison.
4. The manuscript mentioned that the catalyst is intended for use in the cold start phase of motor vehicles. It is suggested to supplement long-term stability tests at low conversion rates (e.g., T50), low-temperature water vapor stability experiments, etc.
5. The authors asserted that PtCu/MgO catalyst forms Pt_xCu_yO_z binary oxide clusters after air calcination. However, the high Cu content (5.3 wt%) compared to Pt raised questions about the existence of separate Cu species or CuOx species. How can this be ruled out?
6. On page 10, line 186, the manuscript mentioned, "the average oxidation state of Cu species is +1.6 and +1.5 for PtCu/MgO (H₂) and Cu/MgO (H₂).". Since the electronegativity of Pt is higher than that of Cu, electrons will transfer from Cu to Pt in the PtCu alloy, lowering Pt's average oxidation state from +0.4 to +0.2. Accordingly, the average oxidation state of Cu should increase after losing electrons. However, the authors attributed the increase in Cu's oxidation state to the formation of isolated CuOx species. The

author should provide an explanation for this.

7. On page 13, line 225, the authors mentioned a more obvious variation in the average oxidation state of Cu species from +1.5 to +2 for both PtCu/MgO (used) and Cu/MgO (used). From Table S3, it is evident that the average oxidation state of Cu in Cu/MgO (used) is +1.8, while in PtCu/MgO (used), it is +2. This discrepancy is not apparent compared to the authors' description in the manuscript. Furthermore, the average oxidation state of Pt in Pt/MgO (used) is +1.0, whereas in PtCu/MgO (used), it is +1.4, highlighting a more pronounced inconsistency with the manuscript's description.

8. On page 13, line 230, the authors attempted to demonstrate that PtCu alloy transforms into PtCu alloy plus CuOx after CO oxidation. The evidence provided by the author did not adequately support this conclusion. Many questions remained regarding the relationship between CuOx species and activity. For instance, did the generation of CuOx species depend on the time and temperature of CO oxidation? Will multiple cycles of CO oxidation tests disrupt the structure of PtCu alloy and CuOx species? The authors should design experiments to prove this and consider supplementing relevant in-situ experiments, such as in-situ EXAFS.

9. The authors repeatedly mentioned the presence of CuOx species on the surface of PtCu alloy in the manuscript. The authors should provide more direct evidence to support this claim.

10. In Table S3, the authors stated that the average oxidation state of Cu in Cu/MgO catalyst decreases from +2.0 to +1.5 under hydrogen conditions. The decrease in oxidation state is slight. Is it due to limitations in reduction time or temperature, preventing complete reduction of Cu to the metallic state?

11. The authors attempted to prove the structural evolution from PtxCu_yO_z cluster to alloy and alloy-clusters interface in the manuscript. However, existing experimental evidence does not support the author's viewpoint. After reduction treatment, the author believes that PtCu alloy is formed, but is there elemental Cu present? After CO oxidation, the authors suggest that PtCu alloy further evolves into PtCu alloy and CuOx cluster structures, but is PtOx cluster species also present, as the XPS data for the used PtCu/MgO catalyst shows an increase in Pt positive oxidation states?

12. In Fig. S13, d and l, the comparison at 2064 cm⁻¹ attributed to PtO-CO characteristic peak suggests that Pt/MgO (used) disappears earlier than PtCu/MgO (used). Can this imply that CO poisoning is more pronounced in PtCu/MgO (used), contradicting the manuscript's claim that PtCu/MgO (used) is more resistant to CO poisoning? It is recommended to provide more convincing evidence that PtCu alloy protects Pt sites from CO poisoning.

13. Check the format of Supplementary Figure 2.

14. In Fig. 5c, the metal oxidation states are not indicated, and there are issues with peak assignments.

Response to the Reviewers' Comments

Reviewer #1

The manuscript from Yunan Li et al. addresses the CO oxidation reaction mechanism on the Pt-Cu-MgO powder catalyst. By means of XAS, TEM and H₂-TPR, the PtCu catalyst was characterized. Further combined with reactivity data, MS and FT-IR, the CO oxidation reaction mechanism was resolved, which was dependent on the reaction temperature. The MvK reaction mechanism prefers low temperatures in contrast to the L-H mechanism occurring at higher temperatures. The new mechanistic insights into the small PtCu clusters towards CO oxidation reaction and also its predominant reactivity contribute to the research on the advanced catalyst. Although, the manuscript is well written, the results and discussions are not clearly presented. Therefore, I suggest that this manuscript should be suitable for publication in Nature Communications after major revision.

1. Pg 8, Fig.1b-1d: using the STEM-EDS mapping, the Pt and Cu species were demonstrated to be well distributed on the MgO surface. However, no higher resolution TEM data were shown. In contrast, the TEM data of the used catalyst indicated the nanostructure of PtCu were ~2nm in diameters (Fig. 4). Was the sintering taking place during the activation or reaction? The authors need to addressed this structural change with more microscopic data. A follow-up question regarding this Fig.1h-1i, the XAS (both NEXAFS and EXAFS) data indicated the similarity between CuPt-MgO and Cu-MgO. Considering the high loading of Cu (6wt%), why no Cu clusters were observed in the Fig.1b-1c. Besides, as the structural diversity of this PtCuMgO may post great challenge for the EXAFS fitting, how the authors exclude those of Pt clusters and Cu clusters, and only confirm the existence of Pt_xCu_yO_z clusters.

Response: Thanks for the reviewer's valuable comment and suggestion. In order to provide more microscopic data and clearly observe the precise distribution of Pt and Cu elements in PtCu/MgO, HAADF-STEM images and the related mapping were supplied in Fig 1b-d. It can be seen that Pt and Cu element was distributed uniformly at the cluster level within the same areas on the surface of MgO. For the further confirmation about the elemental composition of cluster in PtCu/MgO, the

corresponding line-profile analysis (Fig. 1d) manifest that Pt and Cu homogeneously dispersed throughout the $\text{Pt}_x\text{Cu}_y\text{O}_z$ binary oxide clusters. In addition, because of high loading of Cu species about 6 wt.%, a part of Cu species existed in isolation without forming interaction with Pt species according to the reduction peak at 209 °C (CuO_x cluster) in H_2 -TPR results of Fig. 1e. **Therefore, CuO_x and $\text{Pt}_x\text{Cu}_y\text{O}_z$ cluster coexisted in PtCu/MgO.** However, it is difficult to visually observe Cu clusters in Cu/MgO because of the high dispersion and low contrast between Cu ($Z = 29$) and Mg ($Z = 12$). For the comparison, no visible CuO_x clusters can be observed in the STEM and EDS mapping results of Cu/MgO (Supplementary, Fig. S3). The observed clusters in PtCu/MgO can be attributed to $\text{Pt}_x\text{Cu}_y\text{O}_z$ binary oxide clusters with an average particle size of 1.4 ± 0.3 nm for $\text{Pt}_x\text{Cu}_y\text{O}_z$ clusters. After H_2 activation, there was a structural evolution from $\text{Pt}_x\text{Cu}_y\text{O}_z$ clusters to PtCu alloy with the average size distribution of $1.8 \text{ nm} \pm 0.5 \text{ nm}$ due to slight sintering over active site (Fig. 3a). According to the fitting results of EXAFS of PtCu/MgO in Fig.1g and Supplementary Table S2, the strong Pt-O ($R \approx 1.98 \text{ \AA}$ and $CN \approx 5.5$), Pt-O-Cu ($R \approx 3.11 \text{ \AA}$ and $CN \approx 2.8$), and Pt-O-O shells was acquired. The EXAFS fitting results of different shell is clearly shown in Supplementary Fig. S19 and Table S2. The absence of Pt-O-Pt shells suggests that there is no isolated PtO_x cluster in PtCu/MgO. Furthermore, after H_2 activation, only metallic Pt-Cu shell ($R \approx 2.60 \text{ \AA}$ and $CN \approx 11.8$) can be fitted in the in-situ EXAFS results without metallic Pt-Pt shell (Fig. 3f). Thus, no experiment evidence can demonstrate the existence of PtO_x clusters.

Fig. 1. Structural characterization of PtCu/MgO catalysts. (a) XRD pattern, (b) HAADF-STEM images of PtCu/MgO, (c) EDS mapping results of PtCu/MgO, (d) line-scanning results of PtCu/MgO, (e) H₂-TPR profiles, (f) Pt L₃-edge XANES profiles, (g) Pt L₃-edge EXAFS fitting results (the data are K₂-weighted and not phase-corrected), (h) Cu K-edge XANES profiles, (i) Cu K-edge EXAFS fitting results in R space (the data are K₃-weighted and not phase-corrected), (j) Schematic demonstration of platinum-copper oxide cluster.

Supplementary Fig. S3. EDS mapping results of Cu/MgO catalyst.

Fig. 3. Structural characterization of PtCu/MgO after hydrogen reduction. (a) HAADF-STEM images.

Supplementary Fig. S19. Pt L₃-edge EXAFS profiles of PtCu/MgO and the standard Paths of Pt-O₁, Pt-Cu, Pt-O₂ from PtO₂, CuPtO₂, and PtO₂.

Supplementary Table S2 Averaged oxidation state of platinum (δ) and the corresponding EXAFS fitting results for Pt L₃ edges (R : distance; CN: coordination number; σ^2 : Debye-Waller factor; ΔE_0 : inner potential correction) of PtCu/MgO catalysts.

Sample	δ^a	Scatter	R(Å)	CN	σ^2	ΔE_0 (eV)
PtCu/MgO	3.8	O ₁	1.98±0.01	5.5±0.5	0.003	8.4±1.7
		Cu	3.11±0.02	2.8±0.8	0.005	
		O ₂	3.67±0.02	10.5±2.8	0.003	
Pt/MgO	4	O ₁	2.02±0.01	5.4±0.2	0.003	13.1±0.6
		O ₂	3.10±0.01	6.4±0.5	0.003	
		Pt ₁	3.35±0.01	5.6±0.8	0.008	
		Pt ₂	3.85±0.01	11.0±1.7	0.008	
Pt/MgO(used)	1.0	O	2.02±0.03	1.7±0.4	0.002	11.7±3.7
		Pt	2.77±0.03	6.0±0.9	0.005	
		Pt ₁	3.29±0.07	2.6±1.3	0.007	

^aDetermined by linear combination analysis on the XANES profiles with references of Pt foil ($\delta = 0$)/PtO₂ ($\delta = 4$) for Pt L₃ edges. S_0^2 was fixed at 0.88. The σ^2 values were constrained in order to decrease the number of fit parameters and the correlations between them. The distances for Pt-O, Pt-Cu, Pt-Pt are from the crystal structure of PtO₂, CuPtO₂ and Pt.

Fig. 3. Structural characterization of PtCu/MgO after hydrogen reduction. (f) in-situ Pt L₃-edge EXAFS profiles (the data are K₂-weighted and not phase-corrected).

2. Pg 9, Fig.2: as the long-term catalytic reactivity testing was conduct at 150 C degrees, what is the advantage of the Pt-Cu compared of that of the pure Pt, which seems to be also very active at that temperature? In Fig.4, the PtCu alloy structure is confirmed by the HR-TEM where only 0.01nm difference exists. Do the authors have other evidences to confirm the Pt and Cu were indeed alloyed. As the Cu ratio is much higher than that of the Pt, why the EDX spectra in Fig.4c shows the Cu ratio is much lower than that of the Pt? The TEM data in this whole manuscript need to be polished as useful information is hardly be extracted.

Response: We greatly appreciate the reviewer's comment. The practical application of CO oxidation is to convert environmentally harmful CO gas in automobile exhaust under high gas hourly space velocity. According to the "150°C Challenge" from U.S. Department of Energy (M. Zammit et al., "Future automotive aftertreatment solutions: The 150 °C Challenge workshop report," (U.S. DRIVE Report, Southfield, MI, 2013)), the catalysts should steadily be operated at and below 150 °C with high gas hourly space velocity (190,000 ml·h⁻¹·g_{cat}⁻¹). Therefore, the stability test for PtCu/MgO was conducted at 150 °C maintaining a CO conversion values about 93% without obvious deactivation for 20 h. Meanwhile, pure Pt catalyst also showed a good stability at 150 °C only with a poor conversion efficiency about 20% under same test condition (Fig. 2d).

In order to confirm the Pt and Cu were indeed alloyed, the *quasi* in-situ synchrotron XRD and in-situ XAFS were conducted. The *quasi* in-situ synchrotron XRD pattern of PtCu/MgO(used) in Fig. 4c shows the diffraction signal at 40.1° from (111) crystal plane of PtCu alloy, confirming the existence of PtCu alloy in PtCu/MgO after CO oxidation reaction. Furthermore, the structure-sensitive in-situ XAFS technique was employed to determine the precise structural evolution of active site in Fig. 4e-4j. During the CO oxidation reaction at 90 °C, the Pt-Cu shell remained stable and without obvious the formation of Pt-O shell (Fig. 4f). As the reaction temperature increased to 150 °C, the Pt species was slightly oxidized (+0.5) with the appearance of Pt-O shells. However, the main PtCu alloy was still stabilized. Meanwhile, the ratio of Pt and Cu in PtCu alloy is about 1:1 and the updated line scan

results in Fig. 4b also exhibited the similar signal intensity of Cu and Pt element in PtCu alloy.

Fig. 2. (d) Stability test of PtCu/MgO and Pt/MgO at 150 °C for CO oxidation reaction (GHSV: $190,000 \text{ ml} \cdot \text{h}^{-1} \cdot \text{g}_{\text{cat}}^{-1}$)

Fig. 4. Structural characterization of PtCu/MgO catalysts during CO oxidation (a) aberration-corrected HAADF-STEM images of PtCu/MgO after catalytic CO oxidation, (b) the line scan results

of PtCu/MgO after catalytic CO oxidation, (c) synchrotron XRD pattern, (d) Schematic illustration of CuO_x/PtCu Alloy. (e) in-situ Pt L3-edge XANES profiles, (f) in-situ Pt L3-edge EXAFS profiles (the data are K₂-weighted and not phase-corrected), (g) the average oxidation state of Pt in PtCu/MgO catalysts from XANES spectra, (h) in-situ Cu K-edge XANES profiles, (i) in-situ Cu K-edge EXAFS profiles (the data are K₃-weighted and not phase-corrected), (j) the average oxidation state of Cu in PtCu/MgO catalysts from XANES spectra.

3. Page.14, Fig.5, The XPS data for the Pt element should not be fitted so casually. One specific question about the EXAFS data fitting at Fig.5e, how the author confirmed the peak at ~2.1Å corresponded to the Pt-Cu bond? What does the standard XAS spectra of PtCu alloy look like?

Response: We greatly appreciate the reviewer's comment. The XPS data for Pt element has been refitted in Supplementary Fig. S10. The spectra have been corrected based on the binding energy of C 1s neutral carbon peak at 284.8 eV. XPS spectra for PtCu/MgO (used) were deconvoluted into two doublet lines, each being represented by a pair of symmetric spin-orbit components, were related to metallic platinum (binding energy of BE (Pt 4f_{7/2}) = 71.7 eV) and platinum oxides (PtO ~74.1 eV). The Fourier transforms spectrum in the R space from the k₂-weighted EXAFS of PtCu/MgO(used) exhibits the peak at ~2.60 Å (Supplementary Fig. S20), which is significantly different from the Pt foil, and is the same as the standard path of Pt-Cu from CuPt (mp-644311 in The Materials Project, Annalen der Physik (Leipzig) 82, 449–478 (1927)). It proves that PtCu alloy structure is stable during the whole CO oxidation reaction. Moreover, the *quasi* in-situ synchrotron XRD pattern of PtCu/MgO(used) in Fig. 4c also shows the diffraction signal at 40.1° from (111) crystal plane of PtCu alloy, confirming the existence of bimetallic PtCu phase for PtCu/MgO catalysts after CO oxidation reaction.

Supplementary Fig. S10. XPS spectra of Pt 4f for PtCu/MgO after catalytic CO oxidation.

Supplementary Fig. S20. Pt L_3 -edge EXAFS profiles of PtCu/MgO after CO oxidation reaction, Pt foil and the standard paths of Pt-Cu from CuPt.

Fig. 4. (c) *quasi* in-situ synchrotron XRD (SXRD) pattern of PtCu/MgO after CO oxidation reaction.

4. A big question is about the characterization of the used-catalyst, for example, the EPR of the used PtCu catalyst, why this characterization is not conducted on the activated PtCu catalyst? Follow-up question is about the H₂-TPR in Fig.6, this 12C degree difference in the reaction between H₂ and the lattice oxygen seems not to be efficient to explain the big difference in the light-off curve of the CO oxidation reaction in the Fig.2b, where roughly ~80C degrees gap existed for the same ~20% conversion.

Response: We thank reviewer's valuable comments and suggestions. The EPR of all catalysts after H₂ activation in Fig. R1 exhibited symmetric signals at g of about 2.003, attributed to electrons trapped on the O_v. There is a similar trend of EPR signals between activated and used catalysts. It was also found that PtCu/MgO(H₂) possessed the most abundant oxygen defects, which may be beneficial to good CO oxidation activity.

Conventionally, it is good way to investigate the reducibility and interaction for active sites over supported catalysts in H₂-TPR experiments. For the H₂-TPR experiments for Cu/MgO(used) and PtCu/MgO(used), there is an obvious reduction peak at 170 or 155 °C, which indicated that the CuO_x species on the surface of PtCu alloy in PtCu/MgO is more reducible than the conventional CuO_x species in Cu/MgO. There is no necessary connection between the reducibility and CO oxidation. In

the revised version, we have deleted the inappropriate description of “which can explain the difference in activity between PtCu/MgO and Cu/MgO”. In addition, the active oxygen species can be considered for the promotion of CO oxidation activity. In Fig. 5c, the in-situ CO-TPR results indicated a huge difference in the CO₂ generation temperature between PtCu/MgO (50 °C) and Cu/MgO (150 °C), which can explain the difference in activity between PtCu/MgO and Cu/MgO. In the revised version, the description about in-situ CO-TPR has been updated: “The CuO_x species on the surface of PtCu alloy may motivate initial CO oxidation (~50 °C) through Mars-van Krevelen (MvK) mechanism, promoting the CO oxidation activity.” in Line 13-14, Page15.

Fig. R1. EPR spectra of PtCu/MgO catalyst after hydrogen reduction.

Fig. 5. The reducibility and active oxygen for PtCu/MgO catalysts. (a) In-situ H₂-TPR profiles for used PtCu/MgO catalysts after CO oxidation without contact to air, (b) XPS spectra of Cu 2p for used PtCu/MgO catalysts, (c) CO-TPR patterns of PtCu/MgO catalysts, (d) EPR spectra of used PtCu/MgO catalysts.

5. More in-situ data need to provide to support the schematic illustrations in Fig.8, for example before and after H₂ activation, before, during, and after the CO oxidation reaction.

Response: We thank reviewer's valuable comments and suggestions. In-situ XAFS experiments for PtCu/MgO before and after H₂ activation, during CO oxidation reaction and quasi in-situ synchrotron XRD for PtCu/MgO after H₂ activation and CO oxidation reaction were carried out to support the schematic illustrations in Fig.8. The corresponding description was added in the revised manuscript in Line 14, Page 9 to Line13, Page 10 and Line 5, Page 12 to Line 12 Page 13. **"In order to further confirm the formation of PtCu alloy, the quasi-situ synchrotron XRD and in-situ XAFS were conducted. The XRD pattern for PtCu/MgO after hydrogen reduction exhibited two additional peaks positioned at 40.1°, 50.4° (Fig. 3e), which match with the (111) planes of PtCu alloy and the (200) planes of metallic Cu, respectively. The in-situ XANES in Supplementary Fig. S6, and Fig. S7 shows that the average oxidation state of Cu and Pt were both 0 valence in PtCu/MgO after hydrogen reduction. The in-situ EXAFS of Pt L₃-edge for PtCu/MgO(H₂) exhibits one prominent peak at ~2.60 Å (Fig. 4f) with a coordination number of ~11.8 (Supplementary, Table S6), which is significantly different from the Pt foil, and is in accordance with the standard path of Pt-Cu from CuPt (mp-644311 in The Materials Project)³⁹. The in-situ EXAFS of Cu K-edge for PtCu/MgO(H₂) in Fig. 3g exhibits one prominent peak of Cu-Cu shell at ~2.59 Å (CN ≈ 6.3), which indicating that copper oxide clusters were reduced into metallic Cu particles after hydrogen pretreatment. To further strengthen the formation of PtCu alloy,**

the wavelet transforms (WT) analysis of Pt EXAFS oscillations was conducted. As illustrated by the WT contour plots of Pt foil (Fig. 4h) and PtO₂ (Supplementary Fig. S8), the intensity maxima at ~10 Å⁻¹ and ~5 Å⁻¹ are attributed to the Pt-Pt and Pt-O contributions, respectively. In contrast, for the WT contour plot of PtCu/MgO(H₂) (Fig. 4i), one intensity maximum at near 7 Å⁻¹ is exclusively observed, which is assigned to the Pt-Cu contribution. Therefore, the results of HAADF-STEM, quasi in-situ synchrotron XRD and in situ XAFS characterizations verify the formation of PtCu alloy in PtCu/MgO after H₂ reduction.”

“The quasi in-situ synchrotron XRD pattern of PtCu/MgO(used) in Fig. 4c also shows the diffraction signal at 40.1° from (111) crystal plane of PtCu alloy, confirming the existence of PtCu alloy in PtCu/MgO after CO oxidation reaction. For MgO supported Pt sample (Supplementary, Fig. S9), the main active site is metallic platinum with 0.23 nm for Pt (111) plane. Furthermore, the structure-sensitive in-situ XAFS technique was employed to determine the precise structural evolution of active site in Fig. 5e-5j. During the CO oxidation reaction at 90 °C, the Pt-Cu shell remained stable and without obvious the formation of Pt-O shell (Fig. 4f). As the reaction temperature increased to 150 °C, the Pt species was slightly oxidized (+0.5) with the appearance of Pt-O shells, well consistent with the XPS results in Supplementary Fig. S10. However, the main PtCu alloy was still stabilized. For Pt/MgO catalysts after CO oxidation reaction, a major metallic Pt-Pt shell ($R \approx 2.77 \text{ \AA}$, $CN \approx 6.0$) plus Pt-O shell ($R \approx 2.02 \text{ \AA}$, $CN \approx 1.7$) can be fitted (Supplementary, Fig S11 and Table S2), indicating the existence of PtO_x species was not the dominating factor in promoted CO oxidation activity. For Cu species in PtCu/MgO, it can be reduced to metallic Cu species with zero valence after hydrogen reduction at 500 °C. When it switched to CO oxidation condition at 90 °C, Cu-O and Cu-O-Cu shells began to appear, indicating the formation of CuO_x species (Fig. 4i). During the CO oxidation reaction

at 150 °C, the metallic Cu-Cu shell completely disappeared with an oxidation state of +0.4, indicating that the reoxidation and redispersion of isolated Cu species and Cu species on the surface of PtCu alloy. For comparison, the XAFS data for Cu/MgO were also acquired in Supplementary Fig. S12. The isolated CuO_x species can be reduced to metallic Cu component after hydrogen reduction and remained at metallic state (Cu⁰) during the CO oxidation at 90 °C. It indicated that in PtCu/MgO sample after hydrogen activation, when the reaction gas was switched into reactor, the Cu atoms in PtCu alloy could rapidly disassociate O₂ gas into oxygen species, together with the formation of CuO_x species on the surface of PtCu alloy. Furthermore, the metallic Cu particles can be seen in Cu/MgO (used) sample by the aberration corrected HAADF-STEM (Supplementary, Fig. S9), and there is still small Cu-Cu shell (CN ≈ 0.7) can be fitted in Cu/MgO (used) (Supplementary, Fig. S13 and Table S3), indicating that the complete dispersion of the isolated metallic Cu particles into small CuO_x clusters requires higher temperature in Cu/MgO. Tomita et al. also reported that the formation of the interface between metallic Pt nanoparticles and FeO_x promoted the oxidation of CO at low temperature¹⁴. Furthermore, the aberration-corrected HAADF-STEM images of PtCu/MgO can also displayed some clusters at subnanometer on the surface of PtCu alloy, which may be assigned to CuO_x cluster on the basis of element contrast (Supplementary, Fig. S9). Therefore, it can be concluded that the main active site for PtCu/MgO during CO oxidation is PtCu alloy with surface CuO_x species (Fig. 4d).”

Fig. 3. Structural characterization of PtCu/MgO after hydrogen reduction. (a) HAADF-STEM images, (b) HRTEM, (c) EDS mapping results of PtCu/MgO after hydrogen pretreatment, (d) Schematic illustration of PtCu Alloy, (e) synchrotron XRD graph, (f) in-situ Pt L₃-edge EXAFS profiles (the data are K₂-weighted and not phase-corrected), (g) in-situ Cu K-edge EXAFS profiles (the data are K₃-weighted and not phase-corrected), (h, i) WT-EXAFS contour plot of Pt L₃-edge signals for Pt foil and the PtCu/MgO after hydrogen reduction.

Fig. 4. Structural characterization of PtCu/MgO catalysts during CO oxidation (a) aberration-corrected HAADF-STEM images of PtCu/MgO after catalytic CO oxidation, (b) the line scan results of PtCu/MgO after catalytic CO oxidation, (c) synchrotron XRD pattern, (d) Schematic illustration of CuO_x/PtCu Alloy. (e) in-situ Pt L3-edge XANES profiles, (f) in-situ Pt L3-edge EXAFS profiles (the data are K_2 -weighted and not phase-corrected), (g) the average oxidation state of Pt in PtCu/MgO catalysts from XANES spectra, (h) in-situ Cu K-edge XANES profiles, (i) in-situ Cu K-edge EXAFS

profiles (the data are K_3 -weighted and not phase-corrected), (j) the average oxidation state of Cu in PtCu/MgO catalysts from XANES spectra.

Reviewer #2

The authors conducted an investigation into PtCu/MgO catalysts for CO oxidation, employing a series of quasi in-situ characterization experiments to elucidate the catalyst's structural changes under reductive and oxidized conditions. Additionally, they explored the reaction mechanism of PtCu-CuOx catalysts across varying reaction temperatures. While the reported data represents a comprehensive investigation, the reviewer has discerned that this work, rather than ushering in novel findings, makes an incremental contribution. Consequently, this manuscript lacks the level of novelty for publication in a prestigious journal like Nature Communications. The PtCu/CuOx catalytic system, the identification of the combined MvK and LH mechanism, and the utilization of approaches such as isotopic labeling are not original. I would suggest its submission to a journal in a more specified field. In addition to the references cited within the manuscript, it is prudent to consider the following related studies:

[1]. "Catalytic activity of PtCu intermetallic compound for CO oxidation: A theoretical insight," published in Catalysis Today, 383, 2022, 339-344.

[2]. "Dual Active Sites of CO Adsorption and Activation over PtCu Alloy Nanocages for Preferential Oxidation of Carbon Monoxide," featured in ACS Applied Energy Materials, 2022, 5(1), 604-614.

[3]. "Ordered intermetallic Pt–Cu nanoparticles for the catalytic CO oxidation reaction," presented in RSC Advances, 2016, 6, 85634-85642.

[4]. "PtCu Intermetallic Compound Supported on Alumina Active for Preferential Oxidation of CO in Hydrogen," found in The Journal of Physical Chemistry C, 2013, 117(20), 10483–10491.

Response: We have read these articles carefully. Ref. [1] provided by reviewer analyzed, from a theoretical point of view, the role of Cu on PtCu intermetallic surfaces in CO oxidation reaction: the CO molecule adsorbs at Pt sites and the Cu atoms remain free for O₂ molecule. Ref. [2] mainly focused the morphology effect of huge PtCu alloy (> 500 nm) on CO-PROX activity and Ref. [3] investigated the effect of ratio for Pt/Cu in intermetallic Pt–Cu nanoparticles on CO oxidation activity. In Ref. [4], Komatsu and co-workers primarily studied the effect of support and preparation method for PtCu intermetallic compounds on PROX reaction. Although these papers, to some extent, studied and reported the catalytic performance of PtCu alloy from different aspects in CO oxidation, there were

few discussions about the detailed and precise structural evolution of PtCu alloy in oxidizing condition and the distinct effects of alloy and oxidized species on CO oxidation activity. Thus, in our work, we have deeply researched the comprehensive structural evolution of PtCu bimetallic system under different condition by the help of a series of in-situ, quasi in-situ characterization method. In this work, our significant findings are as followings:

The platinum copper bimetallic catalyst has excellent catalytic CO oxidation activity and low activation energy. PtCu/MgO catalysts with a low Pt loading weight of 0.5 wt% were prepared by co-impregnation method, and displayed better performance compared to corresponding monometallic catalyst and the reported PtCu/Al₂O₃ and PtCu/TiO₂ in Supplementary Table S5. The apparent activation energy (E_a) of the PtCu/MgO catalyst is around 38 kJ · mol⁻¹, which is much lower than that of Pt/MgO (83 kJ · mol⁻¹), Cu/MgO (59 kJ · mol⁻¹) and other reported E_a values in Fig. S3 (from 42 to 98 kJ · mol⁻¹) of Pt-based³⁰⁻³² or Cu-based catalysts^{8,9}, indicating the difference in active structure or reaction pathway.

Deep investigation of structural evolution over PtCu/MgO catalysts before (Pt_xCu_yO_z binary oxide cluster) and after H₂ activation (PtCu alloy), and during CO oxidation reaction (PtCu alloy with surface CuO_x species). In detail, firstly, the platinum copper bimetal formed Pt_xCu_yO_z binary oxide cluster on MgO support after 400 °C of air calcination. Then, there was an obvious reconstruction of platinum-copper oxide cluster to PtCu alloy after hydrogen reduction. Afterwards, during the CO oxidation reaction, a fraction of Cu atoms on the PtCu alloy surface were oxidized into CuO_x. The simultaneous existence of the PtCu alloy and CuO_x species enables a synergy for catalyzing CO oxidation.

Unraveling the distinct effects of CuO_x and PtCu alloy Sites in CO oxidation. At low temperature, it was discovered that the CO molecule prefers to adsorb on surface CuO_x species, and approximately 63% of the total CO₂ is formed by active oxygen species provided by CuO_x component (M-vK mechanism). At high temperature, both CuO_x and PtCu alloy work together to activate gases oxygen to participate CO oxidation. Around 60% of the acquired CO₂ is produced by O atoms from the introduced O₂ gas adsorbed on PtCu alloy and CuO_x (L-H mechanism).

These literatures provided by reviewer are all valuable and helpful for improving our article. We have added the four references as ref. [18, 19, 20, 25] in the revised version.

Reviewer #3 (Remarks to the Author):

This manuscript has been employed a simple co-impregnation method to prepare PtCu/MgO catalysts. By utilizing various characterization techniques, it was observed that under reduction and oxidation conditions, platinum-copper oxide cluster underwent significant restructuring into PtCu alloy-CuOx interfaces. The synergistic effect between PtCu alloy and CuOx clusters endowed the catalyst with excellent CO oxidation activity. In situ DRIFTS/CO-TPR and isotope labeling experiments indicated that surface CuOx species at low temperatures could stimulate CO oxidation via the M-vK mechanism, while rising to higher temperatures, CO oxidation followed the L-H mechanism. This work holds certain research significance, particularly regarding the structural evolution from platinum-copper oxide cluster to alloy and alloy with cluster, providing important insights into active sites. However, the manuscript also exhibits notable shortcomings. For example, the innovative contributions of the manuscript are not explicitly delineated, and there is a lack of more direct evidence for the characterization of active sites. Additionally, many analytical aspects within the manuscript suffer from internal contradictions. Consequently, as outlined below, several important aspects need addressing before recommending this article for publication in Nature Communications. Specific suggestions are as follows:

1. The Introduction lacked logical coherence. While the manuscript discussed Pt-Cu alloy and CuOx interfaces as active sites for CO oxidation, the authors did not sufficiently elaborate on the scientific basis for choosing Pt and Cu as the research subjects in the introduction, especially in explaining the scientific rationale behind the formation of Pt and Cu alloy. Additionally, the manuscript failed to summarize the research progress and existing issues related to PtCu alloy catalysts in CO oxidation. The current manuscript form did not adequately demonstrate the value of studying the PtCu alloy catalyst system. Furthermore, the rationale for selecting the carrier should be explained in the introduction. Lastly, the third paragraph of the mechanism explanation was overly lengthy and should be appropriately condensed.

Response: We thank reviewer's valuable comments and suggestions. According to your nice suggestions, we have made extensive corrections to our introduction of previous draft. The updated

introduction part in the revised manuscript is: “In CO oxidation reaction, there are a lot of methods to improve the catalytic performance of Pt-based catalysts, such as constructing alloy component, building metal-support interaction⁷, metal-oxide⁸ and metal-metal hydroxide⁹ active sites. Among these methods, alloying Pt with another metal to form bimetallic alloy nanoparticles has been proven to be a promising pathway to improve the catalytic properties, such as Pt-Pd¹⁰, Pt-Rh¹¹, Pt-Au¹², Pt-Ag¹³, Pt-Fe¹⁴, Pt-Co¹⁵, Pt-Ni¹⁶, Pt-Cu¹⁷⁻²⁰ alloy. It is well known that the elements with similar electronegativity are more likely to form alloy because of the easy share of electron. Copper (Cu), as a representative transition metal, has received wide research due to various valence state and rich surface oxygen species^{21,22}. From the perspective of alloy, Cu is a suitable element with a relatively narrow electronegativity gap and atomic radius difference with Pt atom, contributing to the substitution in the lattice and avoiding significant lattice distortion²³. Furthermore, PtCu alloy are also revealed as a promising catalyst in CO oxidation due to low cost, special geometric, electronic and multifunctional effects. Komatsu et al.²⁴ found that PtCu/SiO₂ alloy catalysts exhibited a better CO oxidation activity because of electron transferring from Pt to Cu to promote the activation of O₂. Furthermore, Liu et al.²⁵ exploited that metallic Pt and Cu⁺ in PtCu nanocage alloy could provide dual active sites for the adsorption of CO and O₂. In addition, PtCu alloy samples were employed as high-performance catalysts in other oxidation reactions, such as, methanol oxidation²⁶, selective catalytic oxidation of ammonia²⁷, and polyhydric alcohol oxidation²⁸. Although some detailed and profound researches on PtCu alloy have been investigated, there were still some debatable and unsolved problems for PtCu alloy in CO oxidation. Firstly, if the PtCu alloy can be maintained during the whole CO oxidation? Secondly, the precise dynamic structural evolution of PtCu alloy in oxidation atmosphere such as the oxidation level and the distinct role of Cu and Pt species in consideration of instability and oxygen-

sensitivity of metallic Cu species? Thirdly, the lacking of advanced in-situ characterization method to explore what is the dominant mechanism (MvK or LH) in CO oxidation reaction for PtCu alloy catalysts with various active sites (alloy and oxide species)? The above existed problems matter the exploration and determination of active site and the “structure-activity” relationship in CO oxidation.

For supported catalysts, the choice of support frequently makes an influence on the catalytic performance through the formation of interaction between metal and support to improve the dispersion or regulate the electronic structure of active site. MgO nanosheet offers a combination of high surface area, thermal stability, and cost-effectiveness, making it a favorable catalyst carrier for CO oxidation reaction^{29,30}. Thus, in this work, MgO nanosheet was chosen as a good carrier to prepare platinum copper bimetallic catalysts by co-impregnation method, which displayed good performance compared to monometallic Pt/MgO or Cu/MgO catalyst. The in-situ XAFS results unravel the dynamic structural evolution of active site from platinum-copper oxide cluster to PtCu alloy-CuO_x interface undergoing reductive and oxidized conditions. In situ DRIFTS/CO-TPR and isotope labeling experiments indicated that the CO oxidation can be motivated at ~ 50 °C on surface CuO_x species through M-vK mechanism, in which CuO_x can provide abundant active oxygen species. As the increase of reaction temperature, a moderate CO adsorption on PtCu alloy guarantees enough sites for the activation of gases oxygen into active oxygen species to promote CO oxidation by L-H mechanism.”

2. In Figure 1e, the H₂ reduction peaks of PtCu/MgO catalyst overlapped with the reduction peaks of Cu/MgO and Pt/MgO catalysts. However, the authors attributed this solely to Pt-[Ox]-Cu species. How can the presence of CuO_x or PtO_x species be ruled out?

Response: We greatly appreciate the reviewer’s comment. The H₂-TPR profile of PtCu/MgO was

fitted on the basis of Gaussian distribution. Actually, there is an overlap of reduction peak at 187 °C for Pt-[O]_x-Cu with the peak of CuO_x at 209 °C. Thus, we can confirm the existence of isolated CuO_x species (fig.1e). In addition, according to the EXAFS fitting results of PtCu/MgO, the absence of Pt-O-Pt shell suggests that there is no isolated PtO_x cluster in PtCu/MgO in Fig.1g and Supplementary Table S2. Furthermore, after H₂ activation, only metallic Pt-Cu shell ($R \approx 2.60 \text{ \AA}$ and $CN \approx 11.8$) can be fitted in the in-situ EXAFS results without metallic Pt-Pt shell (Fig. 3f). Thus, no experiment evidence can demonstrate the existence of PtO_x clusters. In the revised version, the picture of H₂-TPR results has been updated and the related description has been corrected in Line 1-3, Page 6: “In addition, because of high loading of Cu species about 6 wt.%, a part of Cu species existed isolatedly without forming interaction with Pt species according to the reduction peak at 209 °C (CuO_x cluster) in H₂-TPR profile of PtCu/MgO (Fig. 1e). Therefore, CuO_x and Pt_xCu_yO_z cluster coexisted in PtCu/MgO.” and Line 14-17, Page 6: “According to the fitting results of EXAFS of PtCu/MgO in Fig.1g and Table S2, the strong Pt-O ($R \approx 1.98 \text{ \AA}$ and $CN \approx 5.5$), Pt-O-Cu ($R \approx 3.11 \text{ \AA}$ and $CN \approx 2.8$), and Pt-O-O shells was acquired, further evidencing the formation of Pt_xCu_yO_z component. The absence of Pt-O-Pt shells suggests that there is no isolated PtO_x cluster in PtCu/MgO.”

Fig. 1. Structural characterization of PtCu/MgO catalysts. (a) XRD pattern, (b) HAADF-STEM images, (c) EDS mapping results, (d) line-scanning results of PtCu/MgO, (e) H₂-TPR profiles, (f) Pt L₃-edge XANES profiles, (g) EXAFS fitting results, (h) Cu K-edge XANES profiles, (i) Cu K-edge EXAFS fitting results in R space, (j) Schematic demonstration of platinum-copper oxide cluster.

Supplementary Table S2 Averaged oxidation state of platinum (δ^a) and the corresponding EXAFS fitting results for Pt L₃ edges (R : distance; CN: coordination number; σ^2 : Debye-Waller factor; ΔE_0 : inner potential correction) of PtCu/MgO catalysts.

Smple	δ^a	Scatter	R(Å)	CN	σ^2	ΔE_0 (eV)
PtCu/MgO	3.8	O ₁	1.98±0.01	5.5±0.5	0.003	8.4±1.7
		Cu	3.11±0.02	2.8±0.8	0.005	

		O ₂	3.67±0.02	10.5±2.8	0.003	
Pt/MgO	4	O ₁	2.02±0.01	5.4±0.2	0.003	13.1±0.6
		O ₂	3.10±0.01	6.4±0.5	0.003	
		Pt ₁	3.35±0.01	5.6±0.8	0.008	
		Pt ₂	3.85±0.01	11.0±1.7	0.008	
Pt/MgO(used)	1.0	O	2.02±0.03	1.7±0.4	0.002	11.7±3.7
		Pt	2.77±0.03	6.0±0.9	0.005	
		Pt ₁	3.29±0.07	2.6±1.3	0.007	

^aDetermined by linear combination analysis on the XANES profiles with references of Pt foil ($\delta = 0$)/PtO₂ ($\delta = 4$) for Pt L₃ edges. S₀² was fixed at 0.88. The σ^2 values were constrained in order to decrease the number of fit parameters and the correlations between them. The distances for Pt-O, Pt-Cu, Pt-Pt are from the crystal structure of PtO₂, CuPtO₂ and Pt.

Fig. 3. Structural characterization of PtCu/MgO after hydrogen reduction. (f) in-situ Pt L₃-edge EXAFS profiles (the data are K₂-weighted and not phase-corrected).

3. In Figure 2d, the authors compared the PtCu/MgO catalyst with Cu/CoFe₂O₄, Pt₃Ti/SiO₂, and Pt/CuCrO_x catalysts, which are evidently different from the PtCu/MgO catalyst discussed in the manuscript. It is suggested that the authors choose more representative catalysts for the comparison.

Response: We greatly appreciate the reviewer's comment and suggestion. We have added a table to summarize the recent monometallic and bimetallic catalysts with different support to compare the T_{50%} (the temperature of 50% CO conversion) with our PtCu/MgO catalyst. The table was added in Supplementary Table S5.

Supplementary Table S5 Comparison of the activities over the representative catalysts for the oxidation of CO.

Catalysts	Pt(wt.%)	Gas feed composition	Gas hourly space velocity (GHSV)/mL·g _{cat} ⁻¹ ·h ⁻¹	T _{50%}	Ref.
Pt/CeO ₂	0.5	2vol.% CO/2vol.% O ₂ /Ar	150,000	210 °C	1
Pt/CeO ₂ -Al ₂ O ₃	1	1vol.% CO/1vol. % O ₂ /Ar	200,000	135 °C	2
PtCu/Al ₂ O ₃	0.4	1vol.CO/air	12,000	142 °C	3
PtCu/TiO ₂	1.6	1vol.CO/air	12,000	140 °C	3
PtCu/SiO ₂	0.3	1vol.CO/air	12,000	143 °C	3

1Pt2Bi/SiO ₂	0.9	1 vol. % CO/20 vol. % O ₂ /N ₂	134,000	85 °C	4
PtCu/MgO	0.5	1 vol. % CO/20 vol. % O ₂ /He	120,000	100 °C	This work

4. The manuscript mentioned that the catalyst is intended for use in the cold start phase of motor vehicles. It is suggested to supplement long-term stability tests at low conversion rates (e.g., T₅₀), low-temperature water vapor stability experiments, etc.

Response: Thanks for the reviewer's valuable comment and suggestion. According to your nice suggestion, low-temperature water vapor stability experiments were performed in Supplementary Fig S5. It was found that there is no serious deactivation about 10 hours in PtCu/MgO stability test and PtCu/MgO exhibited similar CO oxidation activity under the existence of water vapor during light off experiments. Moreover, long-term stability tests for PtCu/MgO at low conversion rates (e.g., T_{50%}) were performed, which showed about 52% conversion without any deactivation within 100 h in Supplementary Fig S5(c).

Supplementary Fig. S5. (a) Catalytic performance for PtCu/MgO under the presence and absence of water vapor condition (1 vol.% CO/20% O₂/79% He, 120,000 ml·h⁻¹·g_{cat}⁻¹), (b) Water vapor stability experiments for PtCu/MgO catalysts at 110°C, (c) Stability test of PtCu/MgO at 115°C for CO oxidation reaction.

5. The authors asserted that PtCu/MgO catalyst forms Pt_xCu_yO_z binary oxide clusters after air calcination. However, the high Cu content (5.3 wt%) compared to Pt raised questions about the existence of separate Cu species or CuO_x species. How can this be ruled out?

Response: We greatly appreciate the reviewer's comment. In order to provide more microscopic data and clearly observe the precise distribution of Pt and Cu elements in PtCu/MgO, HAADF-STEM

images and the related mapping were supplied in Fig 1b-d. It can be seen that Pt and Cu element was distributed uniformly at the cluster level within the same areas on the surface of MgO. In addition, because of high loading of Cu species about 6 wt.%, a part of Cu species existed in isolation without forming interaction with Pt species according the reduction peak at 209 °C (CuO_x) in H_2 -TPR profile of PtCu/MgO (Fig. 1e). For the further confirmation about the elemental composition of cluster in PtCu/MgO, the corresponding line-profile analysis (Fig. 1d) manifest that Pt and Cu homogeneously dispersed throughout the $\text{Pt}_x\text{Cu}_y\text{O}_z$ binary oxide clusters. **Therefore, the isolated CuO_x and $\text{Pt}_x\text{Cu}_y\text{O}_z$ cluster coexisted in PtCu/MgO.**

Fig. 1. Structural characterization of PtCu/MgO catalysts. (a) XRD pattern, (b) HAADF-STEM images of PtCu/MgO, (c) EDS mapping results of PtCu/MgO, (d) line-scanning results of PtCu/MgO, (e) H_2 -TPR profiles, (f) Pt L_{3} -edge XANES profiles, (g) Pt L_{3} -edge EXAFS fitting results (the data are

K₂-weighted and not phase-corrected), (h) Cu K-edge XANES profiles, (i) Cu K-edge EXAFS fitting results in R space (the data are K₃-weighted and not phase-corrected), (j) Schematic demonstration of platinum-copper oxide cluster.

6. On page 10, line 186, the manuscript mentioned, "the average oxidation state of Cu species is +1.6 and +1.5 for PtCu/MgO (H2) and Cu/MgO (H2)." Since the electronegativity of Pt is higher than that of Cu, electrons will transfer from Cu to Pt in the PtCu alloy, lowering Pt's average oxidation state from +0.4 to +0.2. Accordingly, the average oxidation state of Cu should increase after losing electrons. However, the authors attributed the increase in Cu's oxidation state to the formation of isolated CuO_x species. The author should provide an explanation for this.

Response: Thanks for the reviewer's valuable comment. Because of the high sensitivity of metallic Cu species to oxygen, the copper species of samples in the *quasi*-in-situ experimental may be oxidized during the test. Therefore, in-situ XAFS experiments for PtCu/MgO were carried out at Pt L₃-edge and Cu K-edge to provide more intrinsic and reliable electronic and coordination information. The in-situ XANES data indicated that after hydrogen reduction at 500 °C, the average oxidation state of Cu species and Pt species are both 0 in PtCu/MgO (Supplementary, Fig. S6 and S7). Furthermore, there is only strong Pt-Cu shell at ~2.60 Å and Cu-Cu shell at ~2.56 Å at Pt L₃ edge and Cu K edge respectively. It can be seen that the Pt and Cu element can be totally reduced to form PtCu alloy. For Cu/MgO sample after hydrogen reduction at 500 °C (Supplementary, Fig. S12), the average oxidation state of Cu species also is 0 according the Cu K-edge derivative XANES spectra. For Pt/MgO sample after hydrogen reduction at 500 °C (Supplementary, Fig. S11), the average oxidation state of Pt species also is near-zero valence. In the previous manuscript, the description of electronic transfer in PtCu alloy and the average oxidation state of Cu species were inaccurate, which was corrected in the revised version: in Line 17, Page 9 to Line 1, Page 10: "The in-situ XANES in Supplementary Fig. S6, and Fig. S7 shows that the average oxidation state of Cu and Pt were both 0 valence in PtCu/MgO after hydrogen reduction." Line 11-13, Page 12: "As the reaction temperature increased to 150 °C, the Pt species was slightly oxidized (+0.5) with the appearance of Pt-O shells, well consistent with the XPS results in Supplementary Fig. S10." Line 17-21, Page 12: "For Cu species in PtCu/MgO, it can be reduced to metallic Cu species with zero valence after hydrogen reduction at 500 °C. When it switched

to CO oxidation condition at 90 °C, Cu-O and Cu-O-Cu shells began to appear, indicating the formation of CuO_x species (Fig. 4i). During the CO oxidation reaction at 150 °C, the metallic Cu-Cu shell completely disappeared with an oxidation state of +0.4, indicating that the reoxidation and redispersion of isolated Cu species and Cu species on the surface of PtCu alloy.”

Supplementary Fig. S6. (a) in-situ Cu K-edge XANES profiles, (b) The Cu K-edge Derivative XANES spectra of PtCu/MgO catalysts and the references, (c) the average oxidation state of Cu in PtCu/MgO catalysts from XANES spectra, (d) in-situ Cu K-edge EXAFS profiles (the data are k^3 -weighted and not phase-corrected).

Supplementary Fig. S7. (a) in-situ Pt L₃-edge XANES profiles, (b) the average oxidation state of Pt in PtCu/MgO catalysts from XANES spectra by linear combination fitting, (c) in-situ Pt L₃-edge EXAFS profiles (the data are k^2 -weighted and not phase-corrected).

Supplementary Fig. S12. (a) in-situ Cu K-edge XANES profiles, (b) The Cu K-edge Derivative

XANES spectra of Cu/MgO catalysts and the references, (c) Cu K-edge EXAFS profiles of Cu/MgO catalysts (The data are K^3 -weighted and not phase-corrected).

Supplementary Fig. S11. (a) Pt L_3 -edge XANES profiles, (b) Pt L_3 -edge EXAFS profiles of Pt/MgO catalyst, (c) Pt L_3 -edge XAFS (points) and curve-fit (line) of PtCu/MgO (used). (The data are K^2 -weighted and not phase-corrected, the curve-fit was generated using the parameters in Table S3).

7. On page 13, line 225, the authors mentioned a more obvious variation in the average oxidation state of Cu species from +1.5 to +2 for both PtCu/MgO (used) and Cu/MgO (used). From Table S3, it is evident that the average oxidation state of Cu in Cu/MgO (used) is +1.8, while in PtCu/MgO (used), it is +2. This discrepancy is not apparent compared to the authors' description in the manuscript. Furthermore, the average oxidation state of Pt in Pt/MgO (used) is +1.0, whereas in PtCu/MgO (used), it is +1.4, highlighting a more pronounced inconsistency with the manuscript's description.

Response: Thanks for the reviewer's valuable comment. To further acquire more precise evolution of active structure, in-situ XAFS was conducted from hydrogen reduction to CO oxidation at different temperature. As shown in Fig. 4f and Supplementary Table S6, during the CO oxidation reaction at 90 °C, the Pt-Cu shell remained stable and without obvious the formation of Pt-O. As the reaction temperature increased to 150 °C, the Pt species was slightly oxidized (+0.5) with the appearance of Pt-O shell. However, the main PtCu alloy was still stabilized. For Cu species, it can be reduced to metallic Cu species with zero valence state after hydrogen reduction at 500 °C. When it switched to CO oxidation atmosphere at 90 °C, Cu-O and Cu-O-Cu shells began to appear with an average oxidation state of Cu about +0.2, indicating the formation of CuO_x clusters. During the CO oxidation reaction at 150 °C, the metallic Cu-Cu shell completely disappeared with an oxidation state of +0.4, indicating that the reoxidation and redispersion of isolated Cu species and Cu species on the surface of PtCu alloy.

During the CO oxidation reaction at 270 °C, the average oxidation state of Cu for PtCu/MgO is +2. For monometallic Pt and Cu catalysts, the average oxidation state of Pt and Cu after CO oxidation is +1 and +2, respectively. The relevant descriptions of oxidation valence states have been modified in the revised draft: Line 11-13, Page 12: “As the reaction temperature increased to 150 °C, the Pt species was slightly oxidized (+0.5) with the appearance of Pt-O shells, well consistent with the XPS results in Supplementary Fig. S10.” Line 17-21, Page 12: “For Cu species in PtCu/MgO, it can be reduced to metallic Cu species with zero valence after hydrogen reduction at 500 °C. When it switched to CO oxidation condition at 90 °C, Cu-O and Cu-O-Cu shells began to appear, indicating the formation of CuO_x species (Fig. 4i). During the CO oxidation reaction at 150 °C, the metallic Cu-Cu shell completely disappeared with an oxidation state of +0.4, indicating that the reoxidation and redispersion of isolated Cu species and Cu species on the surface of PtCu alloy.”

Fig. 4. Structural characterization of PtCu/MgO catalysts during CO oxidation (a) aberration-corrected HAADF-STEM images of PtCu/MgO after catalytic CO oxidation, (b) the line scan results

of PtCu/MgO after catalytic CO oxidation, (c) synchrotron XRD pattern, (d) Schematic illustration of CuO_x/PtCu Alloy. (e) in-situ Pt L₃-edge XANES profiles, (f) in-situ Pt L₃-edge EXAFS profiles (the data are K₂-weighted and not phase-corrected), (g) the average oxidation state of Pt in PtCu/MgO catalysts from XANES spectra, (h) in-situ Cu K-edge XANES profiles, (i) in-situ Cu K-edge EXAFS profiles (the data are K³-weighted and not phase-corrected), (j) the average oxidation state of Cu in PtCu/MgO catalysts from XANES spectra.

Supplementary Table S6 Averaged oxidation state of platinum (δ) and the corresponding EXAFS fitting results for **Pt L₃ edges** (R : distance; CN: coordination number; σ^2 : Debye-Waller factor; ΔE_0 : inner potential correction) of PtCu/MgO catalysts.

Smple	δ^a	Scatter	R(Å)	CN	σ^2	ΔE_0 (eV)
PtCu/MgO (H ₂ atmosphere at 500 °C)	0	Cu	2.61±0.01	11.8±0.8	0.012	5.3±2.2
PtCu/MgO (Reaction at 90 °C)	0.4	Cu	2.58±0.01	10.6±0.6	0.014	-0.6±1.9
PtCu/MgO (Reaction at 150 °C)	0.5	O	2.04±0.01	1.6±0.6	0.003	1.5±0.7
		Cu	2.64±0.01	7.4±1.6	0.013	
PtCu/MgO (Reaction at 210 °C)	1.1	O	2.03±0.03	1.6±0.6	0.003	6.5±2.2
		Cu	2.65±0.01	6.0±1.3	0.013	
PtCu/MgO (Reaction at 270 °C)	1.5	O	2.04±0.01	2.1±0.3	0.001	12.8±3.0
		Cu	2.72±0.01	4.6±1.0	0.013	
		Pt	3.18±0.01	2.7±1.7	0.006	

^a Determined by linear combination analysis on the XANES profiles with references of Pt foil ($\delta = 0$)/PtO₂ ($\delta = 4$) for Pt L₃ edges. S₀² was fixed at 0.87. The σ^2 values were constrained in order to decrease the number of fit parameters and the correlations between them. The distances for Pt-Pt, Pt-O, Pt-Cu are from the crystal structure of Pt foil, PtO₂, and PtCu.

Supplementary Table S7 Averaged oxidation state of platinum (δ) and the corresponding EXAFS fitting results for **Cu K edges** (R : distance; CN: coordination number; σ^2 : Debye-Waller factor; ΔE_0 : inner potential correction) of PtCu/MgO catalysts.

Smple	δ^a	Scatter	R(\AA)	CN	σ^2	ΔE_0 (eV)
PtCu/MgO (H ₂ atmosphere at 500 °C)	0	Cu	2.59±0.02	6.3±0.9	0.008	3.0
PtCu/MgO (Reaction at 90 °C)	0.2	O	1.88±0.04	0.9±0.5	0.003	-9.1
		Cu	2.51±0.02	1.6±0.5	0.007	
		Cu	3.02±0.02	2.3±0.8	0.007	
PtCu/MgO (Reaction at 150 °C)	0.4	O	1.90±0.02	1.5±0.5	0.003	-7.6
		Cu	3.05±0.02	2.3±0.8	0.008	
PtCu/MgO (Reaction at 210 °C)	2	O	1.93±0.02	1.9±0.4	0.003	-4.9
		Cu	3.09±0.03	1.5±0.7	0.008	
PtCu/MgO (Reaction at 270 °C)	2	O	1.92±0.02	1.9±0.4	0.003	-4.8
		Cu	3.09±0.03	1.3±0.7	0.008	

^a Determined by derivative of the XANES profiles with references of Cu foil ($\delta = 0$) /CuO ($\delta = 2$) for Cu K-edges. S_0^2 was fixed at 0.88. The σ^2 values were constrained in order to decrease the number of fit parameters and the correlations between them. The distances for Cu-Cu and Cu-O are from the crystal structure of Cu foil and Cu₂O.

Supplementary Table S2 Averaged oxidation state of platinum (δ) and the corresponding EXAFS fitting results for Pt L₃ edges (R : distance; CN: coordination number; σ^2 : Debye-Waller factor; ΔE_0 : inner potential correction) of PtCu/MgO catalysts.

Smple	δ^a	Scatter	R(\AA)	CN	σ^2	$\Delta E_0(\text{eV})$
PtCu/MgO	3.8	O ₁	1.98±0.01	5.5±0.5	0.003	8.4±1.7
		Cu	3.11±0.02	2.8±0.8	0.005	
		O ₂	3.67±0.02	10.5±2.8	0.003	
Pt/MgO	4	O ₁	2.02±0.01	5.4±0.2	0.003	13.1±0.6
		O ₂	3.10±0.01	6.4±0.5	0.003	
		Pt ₁	3.35±0.01	5.6±0.8	0.008	
		Pt ₂	3.85±0.01	11.0±1.7	0.008	
Pt/MgO(used)	1.0	O	2.02±0.03	1.7±0.4	0.002	11.7±3.7
		Pt	2.77±0.03	6.0±0.9	0.005	
		Pt ₁	3.29±0.07	2.6±1.3	0.007	

^a Determined by linear combination analysis on the XANES profiles with references of Pt foil ($\delta = 0$)/PtO₂ ($\delta = 4$) for Pt L₃ edges. S_0^2 was fixed at 0.88. The σ^2 values were constrained in order to decrease the number of fit parameters and the correlations between them. The distances for Pt-O, Pt-Cu, Pt-Pt are from the crystal structure of PtO₂, CuPtO₂ and Pt.

Supplementary Table S3 Averaged oxidation state of platinum (δ) and the corresponding EXAFS fitting results for Cu k edges (R : distance; CN: coordination number; σ^2 : Debye-Waller factor; ΔE_0 : inner potential correction) of catalysts.

Smple	δ^a	Scatter	R(\AA)	CN	σ^2	ΔE_0 (eV)
Cu/MgO	2	O	1.97 \pm 0.01	2.0 \pm 0.2	0.003	-3.8 \pm 1.0
		Cu ₁	2.84 \pm 0.02	0.5 \pm 0.3	0.006	
		Mg	2.95 \pm 0.03	4.7 \pm 0.7	0.008	
PtCu/MgO	2	O	1.96 \pm 0.01	2.0 \pm 0.2	0.003	-6.1 \pm 1.0
		Cu ₁	2.81 \pm 0.01	0.6 \pm 0.3	0.006	
		Mg	2.93 \pm 0.02	5.4 \pm 0.8	0.008	
Cu/MgO(used)	2	O	1.91 \pm 0.01	2.1 \pm 0.2	0.003	-7.9 \pm 1.3
		Cu	2.55 \pm 0.02	0.7 \pm 0.2	0.005	
		Cu ₁	3.09 \pm 0.01	1.5 \pm 0.3	0.007	

^a Determined by derivative of the XANES profiles with references of Cu foil ($\delta = 0$)/CuO ($\delta = 2$) for Cu K-edges. S_0^2 was fixed at 0.93. The σ^2 values were constrained in order to decrease the number of fit parameters and the correlations between them. The distances for Cu-Cu, Cu-O and Cu-O-Mg are from the crystal structure of Cu foil, CuO and Mg₃CuO₄.

8. On page 13, line 230, the authors attempted to demonstrate that PtCu alloy transforms into PtCu alloy plus CuO_x after CO oxidation. The evidence provided by the author did not adequately support this conclusion. Many questions remained regarding the relationship between CuO_x species and activity. For instance, did the generation of CuO_x species depend on the time and temperature of CO oxidation? Will multiple cycles of CO oxidation tests disrupt the structure of PtCu alloy and CuO_x species? The authors should design experiments to prove this and consider supplementing relevant in-situ experiments, such as in-situ EXAFS.

Response: Thanks for the reviewer's valuable comment and suggestion. In-situ XAFS experiments for PtCu/MgO and *quasi* in-situ synchrotron XRD for PtCu/MgO for H₂ activation and CO oxidation reaction were carried out to investigate the structural evolution of PtCu/MgO catalysts. The XRD pattern for PtCu/MgO after hydrogen reduction exhibits two additional peaks positioned at 40.1°, 50.4° (Fig. 3e) matching with the (111) planes of PtCu alloy and the (200) planes of metallic Cu, respectively, which indicated that the formation of PtCu alloy. Meanwhile, the in-situ EXAFS of PtCu/MgO(H₂) exhibits one prominent peak at ~2.60 Å (Fig. 3f), which is significantly different from the Pt foil, and is in accordance with the standard path of Pt-Cu from CuPt (mp-644311 in The Materials Project, Annalen der Physik (Leipzig) 82, 449–478 (1927)). Supplementary Table S6 lists the fitting results including coordination numbers (CN) and structural parameters, only Pt–Cu coordination at 2.60 Å is identified, indicating the formation of PtCu alloy. To further strengthen this result, the wavelet transforms (WT) analysis of Pt EXAFS oscillations was conducted. As illustrated by the WT contour plots of Pt foil (Fig. 3h) and PtO₂ (Supplementary, Fig. S8), the intensity maxima at ~10 Å⁻¹ and ~5 Å⁻¹ are attributed to the Pt–Pt and Pt–O contributions, respectively. In contrast, for the WT contour plot of PtCu/MgO(H₂) (Fig. 3i), one intensity maximum at near 7 Å⁻¹ is exclusively observed, which is assigned to the Pt–Cu contribution. Therefore, the results of *quasi* in-situ synchrotron XRD and in situ XAFS characterizations verify the formation of PtCu alloy for PtCu/MgO after H₂ reduction.

Furthermore, in-situ XAFS experiments for PtCu/MgO during CO oxidation reaction were shown in Fig. 4e-4j. It was found that during the CO oxidation reaction at 90 °C, Pt-Cu shell remained stable, while Cu-O and Cu-O-Cu shells began to appear, indicating the appearance of CuO_x clusters. During the CO oxidation reaction at 150 °C, the Cu-Cu shell completely disappears, indicating the reoxidation and redispersion of isolated Cu species and Cu species on the surface of PtCu alloy. The *quasi* in-situ

synchrotron XRD pattern of PtCu/MgO(used) in Fig. 4c also shows the diffraction signal at 40.1° from (111) crystal plane of PtCu alloy after CO oxidation reaction.

Because of the existence of PtCu alloy and isolated Cu species in PtCu-MgO, it is important to distinguish the reoxidation of Cu element in PtCu alloy and isolated Cu species. The XAFS data for Cu/MgO were also acquired in Supplementary Fig S12. The isolated CuO_x species can be reduced to metallic Cu component after hydrogen reduction and it remained at metallic state (Cu^0) during the CO oxidation at 90°C . However, the Cu-O shell already appeared in CO oxidation at 90°C for PtCu/MgO, which indicated that the formation of CuO_x clusters during CO oxidation reaction at 90°C is attributed to the oxidation of copper species on the surface of PtCu alloy by O_2 to form PtCu alloy with surface CuO_x species rather than the reoxidation and redispersion of the isolated metallic Cu particle. The isolated metallic Cu species need more high temperature to be re-oxidized and redispersed.

According to the suggestion of reviewer, multiple cycles of CO oxidation tests were performed (Supplementary, Fig.S14). It was found that the catalyst suffered a slight deactivation during 10th cycling test of CO oxidation reaction, which may due to the disruption of PtCu alloy during the long-time suffering of oxidation atmosphere from room temperature to 300°C . In-situ XAFS experiments for PtCu/MgO during the CO oxidation reaction at 210 and 270°C (Supplementary, Fig.S7) also indicating that the structure of PtCu alloy was destroyed in some degree. However, when the catalyst was reactivated by H_2 , the catalytic activity can be recovered to the initial state.

Fig. 3. Structural characterization of PtCu/MgO after hydrogen reduction. (a) HAADF-STEM images, (b) HRTEM, (c) EDS mapping results of PtCu/MgO after hydrogen pretreatment, (d) Schematic illustration of PtCu Alloy, (e) synchrotron XRD graph, (f) in-situ Pt L₃-edge EXAFS profiles (the data are K₂-weighted and not phase-corrected), (g) in-situ Cu K-edge EXAFS profiles (the data are K₃-weighted and not phase-corrected), (h, i) WT-EXAFS contour plot of Pt L₃-edge signals for Pt foil and the PtCu/MgO after hydrogen reduction.

Supplementary Table S6 Averaged oxidation state of platinum (δ) and the corresponding EXAFS fitting results for **Pt L₃ edges** (R : distance; CN: coordination number; σ^2 : Debye-Waller factor; ΔE_0 : inner potential correction) of PtCu/MgO catalysts.

Sample	δ^a	Scatter	$R(\text{\AA})$	CN	σ^2	$\Delta E_0(\text{eV})$
--------	------------	---------	-----------------	----	------------	-------------------------

PtCu/MgO (H ₂ atmosphere at 500 °C)	0	Cu	2.61±0.01	11.8±0.8	0.012	5.3±2.2
PtCu/MgO (Reaction at 90 °C)	0.4	Cu	2.58±0.01	10.6±0.6	0.014	-0.6±1.9
PtCu/MgO (Reaction at 150 °C)	0.5	O	2.04±0.01	1.6±0.6	0.003	1.5±0.7
		Cu	2.64±0.01	7.4±1.6	0.013	
PtCu/MgO (Reaction at 210 °C)	1.1	O	2.03±0.03	1.6±0.6	0.003	6.5±2.2
		Cu	2.65±0.01	6.0±1.3	0.013	
PtCu/MgO (Reaction at 270 °C)	1.5	O	2.04±0.01	2.1±0.3	0.001	12.8±3.0
		Cu	2.72±0.01	4.6±1.0	0.013	
		Pt	3.18±0.01	2.7±1.7	0.006	

^aDetermined by linear combination analysis on the XANES profiles with references of Pt foil ($\delta = 0$)/PtO₂ ($\delta = 4$) for Pt L₃ edges. S_0^2 was fixed at 0.87. The σ^2 values were constrained in order to decrease the number of fit parameters and the correlations between them. The distances for Pt-Pt, Pt-O, Pt-Cu are from the crystal structure of Pt foil, PtO₂, and PtCu.

Supplementary Fig. S8. WT-EXAFS contour plot of Pt L₃-edge signals for PtO₂.

Fig. 4. Structural characterization of PtCu/MgO catalysts during CO oxidation (a) aberration-corrected HAADF-STEM images of PtCu/MgO after catalytic CO oxidation, (b) the line scan results of PtCu/MgO after catalytic CO oxidation, (c) synchrotron XRD pattern, (d) Schematic illustration of CuO_x/PtCu alloy. (e) in-situ Pt L₃-edge XANES profiles, (f) in-situ Pt L₃-edge EXAFS profiles (the data are K^2 -weighted and not phase-corrected), (g) the average oxidation state of Pt in PtCu/MgO catalysts from XANES spectra, (h) in-situ Cu K-edge XANES profiles, (i) in-situ Cu K-edge EXAFS

profiles (the data are K^3 -weighted and not phase-corrected), (j) the average oxidation state of Cu in PtCu/MgO catalysts from XANES spectra.

Supplementary Fig. S12. (a) in-situ Cu K-edge XANES profiles, (b) The Cu K-edge Derivative XANES spectra of Cu/MgO catalysts and the references, (c) Cu K-edge EXAFS profiles of Cu/MgO catalysts (The data are K_3 -weighted and not phase-corrected).

Supplementary Fig. S14. multiple cycles of CO oxidation tests for PtCu/MgO catalysts.

Supplementary Fig. S7. (a) in-situ Pt L₃-edge XANES profiles, (b) the average oxidation state of Pt in PtCu/MgO catalysts from XANES spectra by linear combination fitting, (c) in-situ Pt L₃-edge EXAFS profiles (the data are K²-weighted and not phase-corrected).

9. The authors repeatedly mentioned the presence of CuO_x species on the surface of PtCu alloy in the manuscript. The authors should provide more direct evidence to support this claim.

Response: Thanks for the reviewer's valuable comment and suggestion. In-situ XAFS data for PtCu/MgO from hydrogen reduction to CO oxidation reaction at 90 and 150 °C was acquired. According to the response to question 8, we have clearly represented the detailed and persuasive evidence of the whole structural evolution for active site from hydrogen reduction (PtCu alloy) to CO oxidation (PtCu alloy with surface CuO_x species). In addition, the aberration-corrected HAADF-STEM images of PtCu/MgO (used) also displayed some clusters at subnanometer on the surface of PtCu alloy, which may be CuO_x cluster on the basis of element contrast in Supplementary Fig. S9(c).

Supplementary Fig. S9. (c) Representative aberration-corrected HAADF-STEM images of PtCu/MgO (used).

10. In Table S3, the authors stated that the average oxidation state of Cu in Cu/MgO catalyst decreases from +2.0 to +1.5 under hydrogen conditions. The decrease in oxidation state is slight. Is it due to limitations in reduction time or temperature, preventing complete reduction of Cu to the metallic state?

Response: Thanks for the reviewer's valuable comment. Because of the high sensitivity for metal Cu species to oxygen, there is some inaccurate description about the average oxidation state of Cu element in previous manuscript. According to in-situ XANES data of Cu/MgO after H₂ activation at 500 °C for 30 mins (Supplementary, Fig. S12), the average oxidation state of Cu in Cu/MgO was 0, indicating the total reduction of CuO to metallic Cu.

Supplementary Fig. S12. (a) in-situ Cu K-edge XANES profiles, (b) The Cu K-edge Derivative XANES spectra of Cu/MgO catalysts and the references, (c) Cu K-edge EXAFS profiles of Cu/MgO catalysts (The data are K³-weighted and not phase-corrected).

11. The authors attempted to prove the structural evolution from $Pt_xCu_yO_z$ cluster to alloy and alloy-clusters interface in the manuscript. However, existing experimental evidence does not support the author's viewpoint. After reduction treatment, the author believes that PtCu alloy is formed, but is there elemental Cu present? After CO oxidation, the authors suggest that PtCu alloy further evolves into PtCu alloy and CuO_x cluster structures, but is PtO_x cluster species also present, as the XPS data for the used PtCu/MgO catalyst shows an increase in Pt positive oxidation states?

Response: Thanks for the reviewer's valuable comment. In order to confirm the formation of PtCu alloy, the quasi-situ synchrotron XRD and in-situ XAFS were conducted. The XRD pattern for PtCu/MgO after hydrogen reduction exhibits two additional peaks positioned at 40.1 and 50.4° (Fig. 2a), which match with the (111) planes of PtCu alloy and the (200) planes of metallic Cu from the reduction of $Pt_xCu_yO_z$ and isolated CuO_x clusters, respectively.

According to the response to question 8, PtCu alloy with surface CuO_x species and isolated CuO_x species coexisted in PtCu/MgO after CO oxidation. Meanwhile, the in-situ XANES results at Pt L3 edge also indicated that the average oxidation state of Pt species in PtCu/MgO was +0.4 and +0.5 in CO oxidation at 90 and 150 °C respectively, according linear combination analysis. It means that about 90% of Pt atom was metallic Pt at 90 °C under the CO conversion of ~40%, which indicated PtO_x species is not the key point for CO oxidation activity < 100 °C. In addition, the adsorption of CO also plays a key role in CO oxidation activity. It was found that CO was mainly adsorbed at Cu^{1+} (2103 cm^{-1}) site and Pt^0 sites (Fig. R2). As the increase of reaction temperature (150 °C), about 13% of Pt is PtO_x species, with the main adsorption of CO on Pt^0 (2069 cm^{-1}) in PtCu alloy. Therefore, as the CO oxidation went, partial Pt atom in PtCu alloy inevitably be oxidized into PtO_x species. But, combination of in-situ XAFS, DRIFTS result, **PtO_x species is not the main active site in PtCu/MgO.**

Fig. R2. (a-c) In-situ DRIFTS spectra for the Pt/MgO, Cu/MgO and PtCu/MgO catalysts under CO oxidation reaction from 30 to 130 °C.

12. In Fig. S13, d and l, the comparison at 2064 cm^{-1} attributed to Pt^0 -CO characteristic peak suggests that Pt/MgO (used) disappears earlier than PtCu/MgO (used). Can this imply that CO poisoning is more pronounced in PtCu/MgO (used), contradicting the manuscript's claim that PtCu/MgO (used) is more resistant to CO poisoning? It is recommended to provide more convincing evidence that PtCu alloy protects Pt sites from CO poisoning.

Response: Thanks for the reviewer's valuable comment. For in-situ DRIFTS spectra for PtCu/MgO (used) catalysts, the experimental results do not prove that PtCu alloy protects Pt sites from CO poisoning. We have corrected the relevant description in the revised manuscript. We were really sorry for that we overlooked the details of in-situ DRIFTS spectra.

13. Check the format of Supplementary Figure 2.

Response: Thanks for the reviewer's valuable comment and suggestion. We have corrected Supplementary Fig. S2.

Supplementary Fig. S2. TEM and HRTEM images of catalysts: (a, b) Pt/MgO, (c, d) Cu/MgO, (e, f) PtCu/MgO.

14. In Fig. 5c, the metal oxidation states are not indicated, and there are issues with peak assignments.

Response: Thanks for the reviewer's valuable comment and suggestion. The metal oxidation states of XPS spectra of Pt 4f have been indicated and the peak was reassigned (Supplementary Fig. S10). The spectra have been corrected based on the binding energy of C 1s neutral carbon peak at 284.8 eV. XPS spectra for PtCu/MgO (used) were deconvoluted into two doublet lines, each being represented by a pair of symmetric spin-orbit components, were related to metal platinum (binding energy of BE (Pt 4f_{7/2}) = 71.7 eV) and platinum oxides (PtO ~74.1 eV). We would like to thank the referee again for taking the time to review our manuscript.

Supplementary Fig. S10. XPS spectra of Pt 4f for PtCu/MgO after catalytic CO oxidation.

REVIEWER COMMENTS

Reviewer #1 (Remarks to the Author):

See review in attached document 
Reviewer #2 (Remarks to the Author):

The authors have addressed my comments. The manuscript is now ready for acceptance.

Reviewer #3 (Remarks to the Author):

By means of comprehensive quasi situ characterization technique, the authors found that PtCu/MgO had obvious structural evolution from $Pt_xCu_yO_z$ binary oxide clusters to PtCu alloys with surface CuOx species. The synergistic effect between PtCu alloy and CuOx species enabled a good CO oxidation activity. At the same time, it was found that PtCu/MgO catalyst followed M-vk mechanism and L-H mechanism in the low and high temperature stages of CO oxidation reaction, respectively. There are many studies on CO reaction and the performance of this study on CO reaction is not outstanding, so the authors need to pay more attention to condensing the innovativeness of the work, otherwise the work is not suitable to be published in Nature Communications. Although the work is detailed, there are still some problems that need to be solved. Specific questions are as follows:

1. In Figure 1g, Pt-O-Cu and Pt-O-Pt appear to be in almost the same location. Is their attribution reliable? Or is there another, more distinguishable representation? Otherwise, it is difficult to show that there are no isolated PtOx clusters in PtCu/MgO.
2. As can be seen from Figure 2b, the low temperature activity of Pt/MgO catalyst is superior to that of Cu/MgO catalyst. However, in Figure 2c, the apparent activation energy of Pt/MgO catalyst given by the authors is much higher than that of Cu/MgO catalyst. This is obviously not appropriate. The authors need to explain this result.
3. In Figure S2, what do (200) and (111) marked in the electron microscope image represent respectively? And the scales used in the drawing are not consistent. Please make sure that the diagrams are standardized.
4. As shown in Figure S5b, the authors only conducted a 10-hour experiment for the water resistance test of PtCu/MgO catalyst, which is a very short time. Longer water resistance experiments need to be added.
5. On Page 11, line 190, the authors mentioned in the original, "After CO oxidation, the aberration corrected HAADF-STEM images showed that the average particle sizes of active site for PtCu/MgO (Fig. 4a), Pt/MgO (Supplementary, Fig. S9) were 1.9 ± 0.6 nm, 1.7 ± 0.4 nm, respectively, which indicated that there is no obvious connection between activity and particle size." It is not clear whether the characterized catalyst is a PtCu/MgO catalyst calcined only in air or a PtCu/MgO catalyst after reduction. Because the initial structure of the two catalysts and the size of the PtCu are inconsistent.

6. On Page 12, line 202, fig 5e-5j is not found. Please check it again.

7. Page 17, line 286, the author mentions, "the abundant active oxygen species in CuOx and the better ability to dissociate O₂ molecule." O₂- TPD experiment is a characterization that characterizes the active oxygen species on the surface, and it is recommended that this experiment be supplemented for testimony.

Response to the Reviewers' Comments

Reviewer #1:

The manuscript is significantly improved compared with the initial submission. Many of the issues regarding figure labeling and data clarity have been resolved. References and discussion have been added to support the reasoning behind the authors' arguments, significantly enhancing the clarity of the manuscript, and some less clear illustrations have also been removed or clarified. However, some significant concerns remain.

In the XPS data, I was happy to see the fitted peak shapes have been adjusted for the Pt 4f region and appear to use proper branching ratios and spin orbit splitting. This is a good improvement. The Pt 4f doublet for metallic Pt should also be asymmetric, but in this case one could argue that even when using asymmetric peaks for Pt⁰, a second doublet would clearly be needed to fit the broad Pt 4f spectrum. Unfortunately, the Cu 3p doublet should be present around 75.1 eV - 77.3 eV, maybe shifted a bit by oxidation. It seems likely that the component labeled Pt²⁺ has a significant contribution from CuO_x, but the Cu 3p peaks have not been mentioned.

Response: We feel great thanks for your professional review work on our article. Indeed, the peaks of Pt 4f and Cu 3p overlap significantly at around 75.1 eV – 77.3 eV. We were really sorry for that we ignored the Cu 3p contribution while fitting Pt 4f spectra. Thank you for your reminding. The XPS spectra for PtCu/MgO (used) were re-deconvoluted into three doublet lines, which are attributed to Pt⁰ (71.5-74.9 eV), Pt²⁺ (73.4-76.7 eV), and Cu²⁺ (76.1-79.4 eV), respectively. The area ratio for the two spin orbit peaks (3p_{1/2}:2p_{3/2}) is 1:2. The area ratio for the two spin orbit peaks (4f_{5/2}:4f_{7/2}) is 3:4. We have added the relevant description in the Supporting Information: Page 11: “The Cu 3p contribution should be taken into account when fitting Pt 4f spectra due to the overlapping of the peaks of Cu 3p and Pt 4f. The XPS spectrum for PtCu/MgO(used) was fitted using three doublets: first at 71.5-74.9 eV corresponding to Pt⁰, second at 73.4-76.7 eV corresponding to Pt²⁺ and third at 76.1-79.4 eV corresponding to Cu²⁺.”

Supplementary Fig. S10. XPS spectra of Pt 4f for PtCu/MgO after catalytic CO oxidation.

While significant improvements have been made to the presentation of many figures, some are still somewhat challenging to understand. For example:

In Fig. 1c, the scale bar is so small that I first searched the main text and figure caption again before finding it. When zooming in on the scale bar, it is possible to see that it likely says 10 nm, but the text is still blurry. Scale bars and labels should be clearly readable.

Response: Thanks for the reviewer's valuable comment and suggestion. According to your nice suggestion, we have made the related correction of scale bar in Fig. 1c.

Fig. 1. Structural characterization of PtCu/MgO catalysts. (a) XRD pattern, (b) HAADF-STEM images of PtCu/MgO, (c) EDS mapping results of PtCu/MgO, (d) line-scanning results of PtCu/MgO, (e) H₂-TPR profiles, (f) Pt L₃-edge XANES profiles, (g) Pt L₃-edge EXAFS fitting results (the data are K²-weighted and not phase-corrected), (h) Cu K-edge XANES profiles, (i) Cu K-edge EXAFS fitting results in R space (the data are K³-weighted and not phase-corrected), (j) Schematic demonstration of platinum-copper oxide cluster.

Finally, while significant improvements have been made to the motivation, background, and discussion of the data, some improvement in the text is still needed. For example:

On page 11, lines 190-192, the text states that the average particle sizes of 1.9 nm and 1.7 nm for PtCu/MgO and Pt/MgO indicate that there is no obvious connection between activity and particle size. However, any difference in activity here could be related to the composition, regardless of particle size. To conclusively state that there is no connection between activity and particle size, it is necessary to describe the activity of particles having similar compositions but different sizes.

Response: Thanks for the reviewer's valuable comment and suggestion. The expression in the manuscript "the average particle sizes of 1.9 nm and 1.7 nm for PtCu/MgO and Pt/MgO indicate that there is no obvious connection between activity and particle size" is inappropriate without the comparison of different size for identical composition. In this work, the research focused on the difference of bimetallic PtCu and monometallic Pt, Cu species on CO activity and the precise structure evolution of active site under distinct atmosphere. Therefore, the sentence of "which indicated that there is no obvious connection between activity and particle size." have been deleted in the revised manuscript. We agree that *conclusively state that there is no connection between activity and particle size, it is necessary to describe the activity of particles having similar compositions but different sizes.* Therefore, the related experiments for the synthesis of PtCu alloy with different particle size by regulating calcination temperature (400 to 800 °C) were conducted to investigate the effect of particle size on CO oxidation activity (Fig. R1-R3). According to HAADF-STEM images in Fig. R2, the average particle size of PtCu sites was about 1.9 nm (400 °C), 2.2 nm (600 °C) and 1.3 nm (800 °C). Meanwhile, the related EDS mapping and the line scan results in Fig. R2 confirmed the element distribution of Pt and Cu in PtCu alloy for PtCu/MgO (600 °C calcination) after CO oxidation. From the perspective of CO oxidation activity, PtCu/MgO with the calcination temperature of 400 °C exhibited the best CO conversion performance. The relationship between particle size, $T_{50\%}$ and calcination temperature was showed on Fig.R3. As the calcination temperature increased from 400 to 600 °C, the size of PtCu alloy particles increased, and the catalytic CO oxidation activity decreased, indicating that there is no positive connection between activity and particle size of PtCu alloy for PtCu/MgO catalysts. In this work, we pay more attention to the structural evolution and the distinct role of PtCu alloy and surface CuO_x species, rather than the particle size of active site on activity. Therefore, the preliminary data of different particle size on activity was not exhibited in manuscript and supporting information. It would be interesting to explore this aspect in more detail in the future.

Fig. R1. Catalytic performance for PtCu/MgO catalysts calcined at different temperature (1 vol.% CO/20% O₂/79% He, 120,000 ml·h⁻¹·g_{cat}⁻¹).

Fig. R2 (a) HAADF-STEM image of PtCu/MgO (600°C calcination) after CO oxidation reaction, (b) HAADF-STEM image of PtCu/MgO (800°C calcination) after CO oxidation reaction, (c) line-scanning results, and (d) EDS mapping results of PtCu/MgO (600°C calcination) after CO oxidation reaction.

Fig. R3 The relationship between particle size/ $T_{50\%}$ and calcination temperature.

On page 6, lines 113-115, the text states ‘For PtCu/MgO, it is difficult to acquire the Cu-O-Pt structure due to the abundant Cu species (5.3 wt.%) and the average coordination information characterization of XAFS technology (Fig. 1i).’ In cases where fundamental limitations are introduced by methodology, it is important to clearly describe what limiting factors are present. Although it can be inferred, in this case, it would be good to more clearly describe what is meant by ‘average coordination information characterization’.

Response: Thanks for the reviewer’s valuable comment and suggestion. XAFS is a bulk technique sensitive to all of the forms of an element in a sample. In actual experiment, millions of x-rays being absorbed by millions of atoms, and the average information was collected. In this work, copper is 12 times more abundant than platinum, the average coordination number of Cu-O-Pt will be very low. Low coordination numbers lead to weaker signal intensities. In such scenarios, accurately extracting the structural information of the sample becomes more challenging because interference noise in the data becomes more significant compared to the characteristic signals. In the revised version, the description about the fitting for the Cu-O-Pt coordination shell has been updated: “For PtCu/MgO, it is a challenge to fit the Cu-O-Pt coordination shell at Cu K edge, because XAFS is a bulk technique sensitive to all of the forms of an element in a sample. In practical XAFS experiment, millions of x-rays being absorbed by millions of atoms, and the average information of one element was collected together, such as Cu-O-Cu (CuO_x) and Cu-O-Pt ($\text{Pt}_x\text{Cu}_y\text{O}_z$) in this system. In this work, the isolated CuO_x species is much abundant than $\text{Pt}_x\text{Cu}_y\text{O}_z$ clusters, which results in the dominated CuO coordination structure.” in Page 6. We would like to thank the referee again for taking the time to review our manuscript.

The authors have made significant improvements to the manuscript. However, because fundamental issues

remain in the data analysis, and significant room for improvements in clarity remain, I would recommend that the analysis be carefully reexamined before the work can be published. As the firstround of review emphasized, the primary findings are of interest, and it would be good to see this work eventually published.

Response: We greatly thank reviewer's comments and suggestions for the improvement of this work!

Reviewer #2 (Remarks to the Author):

The authors have addressed my comments. The manuscript is now ready for acceptance.

Response: We thank reviewer's comment.

Reviewer #3 (Remarks to the Author):

By means of comprehensive quasi situ characterization technique, the authors found that PtCu/MgO had obvious structural evolution from $Pt_xCu_yO_z$ binary oxide clusters to PtCu alloys with surface CuO_x species. The synergistic effect between PtCu alloy and CuO_x species enabled a good CO oxidation activity. At the same time, it was found that PtCu/MgO catalyst followed M-vk mechanism and L-H mechanism in the low and high temperature stages of CO oxidation reaction, respectively. There are many studies on CO reaction and the performance of this study on CO reaction is not outstanding, so the authors need to pay more attention to condensing the innovativeness of the work, otherwise the work is not suitable to be published in Nature Communications. Although the work is detailed, there are still some problems that need to be solved. Specific questions are as follows:

Response: We feel great thanks for your professional review work on our article. These comments are all valuable and helpful for improving our article. As the reviewer's mentioned, for a catalyst, demonstrating outstanding catalytic activity is indeed very important. However, for a fundamental reaction, the CO oxidation is more often used to investigate the "structure-activity" relationship for supported catalysts. Moreover, there is an obvious improvement in activity of PtCu/MgO, compared with monometallic Pt and Cu samples. As we all know, **PtCu alloy samples were employed as high-performance catalysts in many oxidation reactions**, such as, methanol oxidation (*Applied Catalysis B: Environmental*, 2015, 174-175, 361-366), ethanol oxidation (*Nano Energy*, 2021, 88, 106307), selective catalytic oxidation of ammonia (*ACS Catal.* 2023, 13, 7178-7188), and alcohol oxidation (*Advanced Science*, 2017, 4, 1600448). **It is significant to deeply investigate the precise structural information of PtCu alloy.** However, bimetallic alloy nanoparticles are prone to changes during activation or reaction. These changes include reconstructions, segregation and oxidative/reductive evolution, all of which reflect in the variety of activity and durability. Although some detailed and profound researches on PtCu alloy have been investigated. However, most of previous researches were still focused on the research paradigm of synthesis-characterization-performance testing, and the cognition of composition for active site. **There are few investigations of the dynamic structural evolution of PtCu bimetallic under distinct conditions of activation and reaction, which frequently results in the misunderstanding of inherent "structure-activity" relationship.** However, in our work, we have deeply researched the comprehensive structural evolution of PtCu bimetallic system under different condition by the help of a series of in-situ and quasi in-situ characterization method. In addition, the distinct effects of alloy and oxidized species on CO oxidation activity were also explored by in-situ DRITTS and $^{18}O_2$ isotope-labeling and pulse experiments. Our study could provide meaningful guidance for the design and optimization of bimetallic and multi-component catalysts.

In this work, our significant findings are concluded as followings:

Deep investigation of structural evolution over PtCu/MgO catalysts before ($\text{Pt}_x\text{Cu}_y\text{O}_z$ binary oxide cluster) and after H_2 activation (PtCu alloy), and during CO oxidation reaction (PtCu alloy with surface CuO_x species). In detail, firstly, the platinum copper bimetal formed $\text{Pt}_x\text{Cu}_y\text{O}_z$ binary oxide cluster on MgO support after 400 °C of air calcination. Then, there was an obvious reconstruction of platinum-copper oxide cluster to PtCu alloy after hydrogen reduction. Afterwards, during the CO oxidation reaction, a fraction of Cu atoms on the PtCu alloy surface were oxidized into CuO_x . The simultaneous existence of the PtCu alloy and CuO_x species enables a synergy for catalyzing CO oxidation.

Unraveling the distinct effects of CuO_x and PtCu alloy Sites in CO oxidation. At low temperature, it was discovered that the CO molecule prefers to adsorb on surface CuO_x species, and approximately 63% of the total CO_2 is formed by active oxygen species provided by CuO_x component (M-vK mechanism). At high temperature, both CuO_x and PtCu alloy work together to activate gases oxygen to participate CO oxidation. Around 60% of the acquired CO_2 is produced by O atoms from the introduced O_2 gas adsorbed on PtCu alloy and CuO_x (L-H mechanism).

Therefore, based on the above analysis, **this work has solved the precise structural evolution of PtCu alloy, which not only applies for CO oxidation, but also provides important guidance and reference value for the structural variation of other bimetallic or multi-metallic catalysts in oxidative reactions.** Meanwhile, this work can satisfy the high-quality and important advances of significance standards of Nature Communications. Thank you very much for your nice help.

1. In Figure 1g, Pt-O-Cu and Pt-O-Pt appear to be in almost the same location. Is their attribution reliable? Or is there another, more distinguishable representation? Otherwise, it is difficult to show that there are no isolated PtOx clusters in PtCu/MgO.

Response: Thanks for the reviewer's valuable comment and suggestion. According to the fitting results of EXAFS of PtCu/MgO and Pt/MgO in Fig.1g, Supplementary Fig. S19 and Table S2, the distance of Pt-O-Cu and Pt-O-Pt coordination shells is about 3.10 Å and 3.37 Å, respectively. The Pt-O-Pt and Pt-O-Cu shells can be well distinguished. Pt L_3 -edge EXAFS profiles of PtCu/MgO, Pt/MgO and the standard paths used for fitting are shown in Supplementary Fig. S19. In the revised manuscript, the picture Fig. 1g has been updated with a more distinguishable representation.

Fig. 1. (g) Pt L₃-edge EXAFS fitting results (the data are K²-weighted and not phase-corrected).

Supplementary Fig. S19. Pt L₃-edge EXAFS profiles of PtCu/MgO, Pt/MgO and the standard Paths.

Supplementary Table S2 Averaged oxidation state of platinum (δ) and the corresponding EXAFS fitting results for Pt L₃ edges (R : distance; CN: coordination number; σ^2 : Debye-Waller factor; ΔE_0 : inner potential correction) of PtCu/MgO catalysts.

Smple	δ^a	Scatter	$R(\text{\AA})$	CN	σ^2	$\Delta E_0(\text{eV})$
-------	------------	---------	-----------------	----	------------	-------------------------

PtCu/MgO	3.8	O ₁	1.98±0.01	5.5±0.5	0.003	8.4±1.7
		Cu	3.10±0.02	2.8±0.8	0.005	
		O ₃	3.66±0.02	10.5±2.9	0.003	
Pt/MgO	4	O ₁	2.02±0.01	5.3±0.3	0.003	14.6±0.6
		O ₂	3.14±0.01	6.8±0.8	0.003	
		Pt ₁	3.37±0.01	5.0±1.4	0.008	
		Pt ₂	3.86±0.01	9.3±3.0	0.008	
Pt/MgO(used)	1.0	O	2.02±0.03	1.7±0.4	0.002	11.7±3.7
		Pt	2.77±0.03	6.0±0.9	0.005	
		Pt ₁	3.29±0.07	2.6±1.3	0.007	

^aDetermined by linear combination analysis on the XANES profiles with references of Pt foil ($\delta = 0$)/PtO₂ ($\delta = 4$) for Pt L₃ edges. S₀² was fixed at 0.88. The σ^2 values were constrained in order to decrease the number of fit parameters and the correlations between them. The distances for Pt-O, Pt-Cu, Pt-Pt are from the crystal structure of PtO₂, CuPtO₂ and Pt.

2. As can be seen from Figure 2b, the low temperature activity of Pt/MgO catalyst is superior to that of Cu/MgO catalyst. However, in Figure 2c, the apparent activation energy of Pt/MgO catalyst given by the authors is much higher than that of Cu/MgO catalyst. This is obviously not appropriate. The authors need to explain this result.

Response: We sincerely appreciate the valuable comments. We retested the apparent activation energy of catalysts, and began to collect data after the temperature stabilized for 30min during the test to ensure the reliability of the collected data. We try to guarantee almost same temperature range of CO conversion rate for Pt/MgO, Cu/MgO and PtCu/MgO to exclude the difference of temperature on apparent activation energy. For Pt/MgO, the activation energy measured this time (82 kJ·mol⁻¹) is similar to the activation energy measured before (83 kJ·mol⁻¹). For Cu/MgO samples, when the temperature range of the test is close to Pt/MgO, the activation energy is 90 kJ·mol⁻¹, which is higher than that (59 kJ·mol⁻¹) measured at relatively higher temperatures. In the revised version, we have made correction to Fig. 2c.

Fig. 2. (c) Arrhenius plots.

3. In Figure S2, what do (200) and (111) marked in the electron microscope image represent respectively? And the scales used in the drawing are not consistent. Please make sure that the diagrams are standardized.

Response: Thanks for the reviewer's valuable comment. According to your nice suggestion, we have made correction to Supplementary Fig. S2.

Supplementary Fig. S2. TEM and HRTEM images of catalysts: (a, b) Pt/MgO, (c, d) Cu/MgO, (e, f) PtCu/MgO.

4. As shown in Figure S5b, the authors only conducted a 10-hour experiment for the water resistance test of PtCu/MgO catalyst, which is a very short time. Longer water resistance experiments need to be added.

Response: Thanks for the reviewer's valuable comment and suggestion. According to your nice suggestion, longer water resistance experiments were performed in Supplementary Fig S5.

Supplementary Fig. S5. (c) Water vapor stability experiments for PtCu/MgO catalysts at 120°C

5. On Page 11, line 190, the authors mentioned in the original, "After CO oxidation, the aberration corrected HAADF-STEM images showed that the average particle sizes of active site for PtCu/MgO (Fig. 4a), Pt/MgO (Supplementary, Fig. S9) were 1.9 ± 0.6 nm, 1.7 ± 0.4 nm, respectively, which indicated that there is no obvious connection between activity and particle size." It is not clear whether the characterized catalyst is a PtCu/MgO catalyst calcined only in air or a PtCu/MgO catalyst after reduction. Because the initial structure of the two catalysts and the size of the PtCu are inconsistent.

Response: Thanks for the reviewer's valuable comment. For PtCu/MgO catalysts after calcined, the average particle size of $Pt_xCu_yO_z$ binary oxide clusters was 1.4 ± 0.3 nm. After H_2 activation, there was a structural evolution from $Pt_xCu_yO_z$ clusters to PtCu alloy with the average size distribution of $1.8 \text{ nm} \pm 0.5 \text{ nm}$ due to slight sintering. After CO oxidation, the average particle sizes of PtCu alloy for PtCu/MgO was 1.9 ± 0.6 nm, which demonstrated that PtCu alloy structure is stable during CO oxidation reaction. As mentioned in previous version, the average particle size of 1.9 ± 0.6 nm (PtCu/MgO), 1.7 ± 0.4 nm (Pt/MgO) was compared under the same condition: after CO oxidation.

6. On Page 12, line 202, fig 5e-5j is not found. Please check it again.

Response: We greatly appreciate the reviewer's comment. "fig 5e-5j" have been corrected to "fig 4e-4j" in the revised manuscript.

7. Page 17, line 286, the author mentions, "the abundant active oxygen species in CuOx and the better ability

to dissociate O₂ molecule." O₂-TPD experiment is a characterization that characterizes the active oxygen species on the surface, and it is recommended that this experiment be supplemented for testimony.

Response: Thanks for the reviewer's valuable comment and suggestion. O₂-TPD experiment was performed to characterize the active oxygen species on the catalysts surface. According to O₂-TPD results (Supplementary Fig. S21), a distinct desorption peak at the range of 100-200 °C can be ascribed to the surface adsorbed oxygen species. Surface adsorbed oxygen species desorbed at low temperature are known as the active species for catalytic oxidation. The desorption temperature for PtCu/MgO is lower than that of Pt/MgO and Cu/MgO, indicating that the adsorbed oxygen is easier to be activated in PtCu/MgO than Pt/MgO and Cu/MgO.

In addition, we designed a O₂ pulse experiment to test the ability of the oxygen decomposition of each catalyst. The catalysts were firstly treated under 5% H₂/He atmosphere at 500 °C for 30 min. Afterward, the catalyst is then cooled to room temperature in H₂ atmosphere. Then the sample was heated from room temperature to 70 °C in He atmosphere, 2% CO/He (30 mL/min) was then introduced to purging until the baseline is stable. Subsequently, O₂ (20 vol.% O₂/He) pulses of 7.5 mL were injected. The signals of CO₂ (m/z = 44), O₂ (m/z=32) were detected. Because of the total reduction of Pt and Cu after H₂ activation, the production of CO₂ is from the decomposition of oxygen gas.

At 70 °C, PtCu/MgO can activate O₂ to generate CO₂, while no obvious CO₂ signal can be seen for Pt/MgO and Cu/MgO due to poor ability to dissociate oxygen gas. As the increase of experiment temperature at 130 °C, the integral area of CO₂ signal for PtCu/MgO catalyst was larger than that for Pt/MgO and Cu/MgO, indicating better decomposition capacity of O₂ for PtCu/MgO. In addition, it was found that there was a CO₂ shoulder peak in PtCu/MgO, which was not consistent with the variation tendency of oxygen gas. It can be attributed to the consumption of oxygen species in CuO_x species with good recyclability on the surface of PtCu alloy. For Pt/MgO and Cu/MgO, the generated CO₂ signal of Pt/MgO was much better than that of Cu/MgO, well consistent with the trend of apparent activation energy. When the O₂ pulse experiment is conducted at 170 °C, there was also additional CO₂ generation for Cu/MgO. However, the peak area of Cu/MgO is smaller than PtCu/MgO, indicating that CuO_x on the surface of PtCu alloy has better oxygen activation ability than isolated Cu species. The corresponding description was added in Page 16-17 of the revised manuscript: **"The ability of oxygen activation. O₂-TPD experiment was performed to characterize the active oxygen species on the catalysts surface. According to O₂-TPD results (Supplementary Fig. S21), a distinct desorption peak at the range of 100-200 °C can be ascribed to the surface adsorbed oxygen species. Surface adsorbed oxygen species desorbed at low temperature are known as the active species for catalytic oxidation. The desorption temperature for PtCu/MgO is lower than that of Pt/MgO and Cu/MgO, indicating**

that the adsorbed oxygen is easier to be activated in PtCu/MgO than Pt/MgO and Cu/MgO. In addition, we designed a O₂ pulse experiment to test the ability of the oxygen decomposition of each catalyst in Fig. 6. At 70 °C, PtCu/MgO can activate O₂ to generate CO₂, while no obvious CO₂ signal can be seen for Pt/MgO and Cu/MgO due to poor ability to dissociate oxygen gas. As the increase of experiment temperature at 130 °C, the integral area of CO₂ signal for PtCu/MgO catalyst was larger than that for Pt/MgO and Cu/MgO, indicating better decomposition capacity of O₂ for PtCu/MgO. In addition, it was found that there was a CO₂ shoulder peak in PtCu/MgO, which was not consistent with the variation tendency of oxygen gas. It can be attributed to the consumption of oxygen species in CuO_x species with good recyclability on the surface of PtCu alloy. For Pt/MgO and Cu/MgO, the generated CO₂ signal of Pt/MgO was much better than that of Cu/MgO, well consistent with the trend of apparent activation energy. When the O₂ pulse experiment is conducted at 170 °C, there was also additional CO₂ generation for Cu/MgO. However, the peak area of Cu/MgO is smaller than PtCu/MgO, indicating that CuO_x on the surface of PtCu alloy has better oxygen activation ability than isolated Cu species.”

Supplementary Fig. S21. O₂-TPD profile of Pt/MgO, Cu/MgO and PtCu/MgO.

Fig. 6. The ability of oxygen activation. O₂ pulse experiment for Pt/MgO (a,d,g), Cu/MgO (b,e,h) and PtCu/MgO (c,f,i) catalysts at different temperature.

REVIEWER COMMENTS

Reviewer #1 (Remarks to the Author):

It is encouraging that the manuscript has improved in clarity and quality with each review cycle. However, after several review cycles, a thorough reading of the manuscript still reveals significant oversights in data analysis and description. For example:

1) On page 8, line 147, the text implies that the activation energies listed in table S4 range from 42-98 kJ/mol. However, the table explicitly lists activation energies of 45- 76 kJ/mol.

2) On page 10, line 172, the text states 'The in-situ EXAFS of Pt L3-edge for PtCu/MgO(H₂) exhibits one prominent peak at ~2.6 Å (Fig. 3f).' However, a quick examination of Fig. 3f reveals that the peak for Pt foil is at ~2.6 Å. The one prominent peak for PtCu/MgO(H₂) is closer to 2.2-2.3 Å.

3) On page 12, line 207, the text states that 'the Pt species was slightly oxidized with the appearance of Pt-O shells, well consistent with the XPS results in Supplementary Fig. S10.' However, precise peak positions for the fitting should be justified by the inclusion of theory, references, or other definitive corroborating information. Additionally, the strong overlap between the peaks labeled Pt²⁺ 4f and the Cu 3p peaks generates very significant uncertainty in the relative peak intensities and positions. Without justification for the choice of Cu 3p peak positions, it is not clear that the Pt²⁺ 4f peaks are even needed. Without such information, the claim about this data implying the presence of oxidized Pt may be unfounded.

Reviewer #3 (Remarks to the Author):

The manuscript elucidates significant structural evolution over PtCu/MgO from Pt_xCu_yO_z binary oxide cluster to PtCu alloy with surface CuO_x species through various in situ and quasi in situ characterization techniques. Compared to the initial submission, the experimental design and mechanism research of this manuscript have been greatly improved. Additionally, the manuscript adequately articulates the novelty and reporting value of this work in response to the feedback received. Therefore, it is agreed to publish this manuscript in Nature Communications.

Reviewer #4 (Remarks to the Author):

Response to the Reviewers' Comments

Reviewer #1:

It is encouraging that the manuscript has improved in clarity and quality with each review cycle. However, after several review cycles, a thorough reading of the manuscript still reveals significant oversights in data analysis and description. For example:

1) On page 8, line 147, the text implies that the activation energies listed in table S4 range from 42-98 kJ/mol. However, the table explicitly lists activation energies of 45- 76 kJ/mol.

Response: We sincerely appreciate the valuable comments. We were really sorry for our careless mistakes. Thank you for your reminding. We have made correction to Supplementary Table S4. In the revised version, the description about the apparent activation energy (E_a) has been updated: “**The apparent activation energy (E_a) of the PtCu/MgO catalyst is around 42 kJ · mol⁻¹, which is much lower than that of Pt/MgO (82 kJ · mol⁻¹), Cu/MgO (90 kJ · mol⁻¹) and other reported E_a values in Supplementary Table S4 (from 45 to 98 kJ · mol⁻¹) of Pt-based^{31,34-36} or Cu-based catalysts^{22,37}”**

Supplementary Table S4 Comparison of the apparent activation energy over the representative Pt-based and Cu-based catalysts for CO oxidation.

Sample	Apparent activation energy (kJ · mol ⁻¹)	Ref.
1 wt% Pt/CeO ₂	45	1
0.7 wt% Pt/SiO ₂	60	2
4 wt% Pt/Al ₂ O ₃	70	3
1 wt% Pt/SiO₂	98	4
20 wt% Cu/TiO ₂	64-76	5
10 wt% Cu/CeO ₂	60-72	6

2) On page 10, line 172, the text states ‘The in-situ EXAFS of Pt L₃-edge for PtCu/MgO(H₂) exhibits one prominent peak at ~2.6 Å (Fig. 3f).’ However, a quick examination of Fig. 3f reveals that the peak for Pt foil is at ~2.6 Å. The one prominent peak for PtCu/MgO(H₂) is closer to 2.2-2.3 Å.

Response: We thank reviewer's comment. All R-space Pt L₃-edge and Cu K-edge EXAFS spectra in this article are the original spectra obtained by Fourier transform of K-space spectra without phase correction. The real coordination distance obtained through fitting with Artemis software is generally the distance shown in the original R-space spectrum plus about 0.3 Å. In Fig. 3f, the horizontal axis is represented by (R+ α) instead of R, implying that the data is not phase corrected. In addition, the annotation for Fig. 3f also describes that **the data are K²-weighted and not phase-corrected**. According to the fitting results of EXAFS of

PtCu/MgO(H₂) in Supplementary Table S6, the coordination distance of the Pt-Cu shell is about 2.6 Å. However, the coordination distance of the Pt-Pt shell for Pt foil is about 2.8 Å.

Fig. 3. Structural characterization of PtCu/MgO after hydrogen reduction. (a) HAADF-STEM images, (b) HRTEM, (c) EDS mapping results of PtCu/MgO after hydrogen pretreatment, (d) Schematic illustration of PtCu Alloy, (e) synchrotron XRD graph, (f) in-situ Pt L₃-edge EXAFS profiles (**the data are K²-weighted and not phase-corrected**), (g) in-situ Cu K-edge EXAFS profiles (the data are K³-weighted and not phase-corrected), (h, i) WT-EXAFS contour plot of Pt L₃-edge signals for Pt foil and the PtCu/MgO after hydrogen reduction.

3) On page 12, line 207, the text states that ‘the Pt species was slightly oxidized with the appearance of Pt-O shells, well consistent with the XPS results in Supplementary Fig. S10.’ However, precise peak positions for the fitting should be justified by the inclusion of theory, references, or other definitive corroborating information. Additionally, the strong overlap between the peaks labeled Pt2+ 4f and the Cu 3p peaks generates very significant uncertainty in the relative peak intensities and positions. Without justification for the choice of Cu 3p peak positions, it is not clear that the Pt2+ 4f peaks are even needed. Without such information, the claim about this data implying the presence of oxidized Pt may be unfounded.

Response: We feel great thanks for your professional review work on our article. The white line (absorption

maximum after the edge) corresponding to 2p to 5d transition of the excited photoelectron in platinum, which is an important indication of the Pt oxidation state. Fig. 4e and Supplementary Fig. S7a shows that the white line peak gradually increased with the increase of reaction temperature, indicating the oxidation of platinum species during CO oxidation reaction. We also calculated averaged oxidation state of platinum for PtCu/MgO catalysts at different reaction temperatures according to linear combination fitting (Supplementary Fig. S7b). The average oxidation state of platinum for PtCu/MgO catalysts during CO oxidation reaction at 150 and 270 °C is close to +0.5 and +1.5, respectively. About 75% Pt species is in the form of Pt²⁺, which evidenced the existence of Pt²⁺ in PtCu/MgO(used) from XANES results. According to the literature (J. Electron. Spectrosc, 2021, 246, 147027, Appl. Surf. Sci., 2023, 614, 156210), the peaks located at 74.9 eV, 75.0 eV and 76.0 eV can be attributed to Cu 3p_{3/2} states of metallic Cu (Cu⁰), Cu₂O (Cu¹⁺) and CuO (Cu²⁺), respectively. The peak at banding energy value of 71.2 eV and 73.6 eV are the 4f_{7/2} orbital of Pt⁰ and Pt²⁺ species, respectively. The Cu 3p contribution should be taken into account when fitting Pt 4f spectra due to the overlapping of the peaks of Cu 3p and Pt 4f. The XPS spectrum for PtCu/MgO(used) was fitted using three doublets: first at 71.5 eV corresponding to 4f_{7/2} orbital of Pt⁰, second at 73.4 eV corresponding to 4f_{7/2} orbital of Pt²⁺ and third at 76.1 eV corresponding to 3p_{3/2} state of Cu²⁺. Therefore, the fitting results of XPS spectrum for PtCu/MgO(used) is reliable. In the revised version, the description has been updated: “Fig. 4e and Supplementary Fig. S7a show that the white line peak gradually increased with the increase of reaction temperature, indicating the slight oxidation of platinum species during CO oxidation reaction with +0.5 and +1.5 state for Pt species at 150 and 270 °C respectively (Fig. 4g and Supplementary Fig. S7). Meanwhile, the XPS results in Supplementary Fig. S10 also evidenced that the platinum species underwent oxidation after CO oxidation.”

Supplementary Fig. S7. (a) in-situ Pt L₃-edge XANES profiles, (b) the average oxidation state of Pt in PtCu/MgO catalysts from XANES spectra by linear combination fitting, (c) in-situ Pt L₃-edge EXAFS profiles (the data are K²-weighted and not phase-corrected).

Reviewer #3:

The manuscript elucidates significant structural evolution over PtCu/MgO from PtxCu_yO_z binary oxide cluster to PtCu alloy with surface CuO_x species through various in situ and quasi in situ characterization techniques. Compared to the initial submission, the experimental design and mechanism research of this manuscript have been greatly improved. Additionally, the manuscript adequately articulates the novelty and reporting value of this work in response to the feedback received. Therefore, it is agreed to publish this manuscript in Nature Communications.

Response: We thank reviewer's comment.

Reviewer #4:

Response: We thank reviewer's comment.

Reviewer #1 (Remarks to the Author):

After several rounds of review, the manuscript is significantly improved. The obvious issues with data interpretation, figures, and discussion that were present in the previous drafts have been corrected, and it is in a state that can be accepted.

Reviewer #4 (Remarks to the Author):

Response to the Reviewers' Comments

Reviewer #1:

After several rounds of review, the manuscript is significantly improved. The obvious issues with data interpretation, figures, and discussion that were present in the previous drafts have been corrected, and it is in a state that can be accepted.

Response: We thank reviewer's comment.

Reviewer #4:

Response: We thank reviewer's comment.